

# Scalable Diagnostics and Data Compression for Global Atmospheric Chemistry using Ristretto Library (version 1.0)

**Meghana Velegar**[1]**, N. Benjamin Erichson**[1]**, Christoph A. Keller**[2,3]**, and J. Nathan Kutz**[1]

[1] Department of Applied Mathematics, University of Washington, Seattle, WA 98195, USA
[2]NASA Global Modelling and Assimilation Office, Goddard Space Flight Center, Greenbelt, MD, USA
[3]Universities Space Research Association, Columbia, MD, USA

**Correspondence:** J. N. Kutz (kutz@uw.edu)

**Abstract.** We introduce a new set of algorithmic tools capable of producing scalable, low-rank decompositions of global spatio-temporal atmospheric chemistry data. By exploiting emerging *randomized linear algebra* algorithms, a suite of decompositions are proposed that extract the dominant features from *big data* sets (i.e. global atmospheric chemistry at longitude, latitude and elevation) with improved interpretability. Importantly, our proposed algorithms scale with the intrinsic rank of the global chemistry space rather than the ever increasing spatio-temporal measurement space, thus allowing for efficient representation and compression of the data. In addition to scalability, two additional innovations are proposed for improved interpretability: (i) a non-negative decomposition of the data for improved interpretability by constraining the chemical space to have only positive expression values (unlike PCA analysis), and (ii) sparse matrix decompositions, which thresholds low-correlations to zero, thus highlighting the dominant, localized spatial activity (again unlike PCA analysis). Our methods are demonstrated on a full year of global chemistry dynamics data, showing its significant improvement in computational speed and interpretability. We show that the here presented decomposition methods successfully extract known major features of atmospheric chemistry, such as summertime surface pollution and biomass burning activities. Indeed, we find that the full annual model output can be reconstructed using only 50-100 principal modes, suggesting that the presented methods offer the potential to archive model data of atmospheric chemistry with compression factors in the range of 200-4000 or greater. In the emerging area of *big data*, specifically global chemistry monitoring, such technologies are critically enabling for real-time and computationally tractable diagnostics of both large scale simulation and measurement data.

## 1 Introduction

Dimensionality reduction is a critically enabling aspect of machine learning and data science in the era of *big data*. Specifically, extracting the dominant low-rank features from a high-dimensional data matrix $\mathbf{X}$ allows one to efficiently perform tasks associated with clustering, classification, reconstruction and prediction (forecasting). Commonly used *linear* dimensionality reduction methods (Cunningham and Ghahramani, 2015) are typically based upon the *singular value decomposition* (SVD) which allows one to exploit covariances manifest in the data. Thus the analysis of big data, such as the atmospheric chemistry data considered here, relies on a variety of matrix decomposition methods which seek to exploit low-rank features exhibited by the high-dimensional data. Despite our ever-increasing computational power, the emergence of large-scale datasets has severely challenged our ability to analyze data using traditional matrix algorithms, especially for ever increasing refinements of computational models. However, we show in this manuscript that innovations that exploit randomness (Halko et al., 2011; Mahoney et al., 2011; Drineas and Mahoney, 2016; Erichson et al., 2016, 2017) have recently been demonstrated as an effective strategy to easing the computational demands of low-rank approximations and data compression from matrix decompositions such as the SVD, thus allowing for a *scalable* architecture and efficient data representation for modern atmospheric chemistry





applications. Moreover, nonnegative matrix factorizations and sparse PCA improve the interpretability of such decompositions, thus providing a mathematical platform to aid in scientific discovery and analysis.

Matrix decompositions are often the workhorse algorithms for scientific computing applications in the areas of applied mathematics, statistical computing, and machine learning. *Principle component analysis* (PCA) is perhaps the most widely used SVD-based method for extracting features from data. It provides an optimal, $\ell_2$ rank-$r$ reconstruction of the data so that

$$\min \|\mathbf{X} - \mathbf{U}\mathbf{U}^T\mathbf{X}\|_2 \tag{1}$$

where $\mathbf{U}$ is the $r$-rank left-singular vectors of the SVD $\mathbf{X} \approx \mathbf{U}\Sigma\mathbf{V}^*$. As surveyed by (Cunningham and Ghahramani, 2015), there are many variants of truncated SVD-based methods, all of which modify the underlying optimization problem associated with producing low-rank modes (features) of the data by imposing constraints on (1). We are specifically concerned with time-series measurements of the concentration of chemical species collected from spatial locations in the atmosphere. On a global scale (longitude, latitude and elevation), this data can be exceptionally high-dimensional so as to be not computationally tractable. Thus computationally scalable methods are required for the analysis of atmospheric chemistry dynamics.

In this work, we present a variety of emerging matrix decomposition methods that can be used for scalable diagnostics and data compression of global atmospheric chemistry dynamics. Atmospheric chemistry is an exceptionally high-dimensional problem as it involves hundreds of chemical species that are coupled with each other via a set of ordinary differential equations. Models of atmospheric chemistry that are used to simulate the spatio-temporal evolution of these chemical constituents need to keep track of each chemical species on a global scale (longitude, latitude, elevation) and at each point in time. The resulting data sets - used for scientific analysis or required for subsequent restarts of the model - quickly become massive, especially as horizontal model resolution steadily increases. For example, a single snap shot of the chemical state of an atmospheric chemistry model at $25 \times 25\,\mathrm{km}^2$ horizontal resolution requires 60 GB of storage space. We use here randomized linear algebra methods (Halko et al., 2011; Mahoney et al., 2011; Drineas and Mahoney, 2016; Erichson et al., 2016) to extract the dominant, low-rank mode structures from a full three-dimensional data set of atmospheric chemistry. These methods are highly scalable and can thus be used on emerging big data sets describing global chemistry dynamics, providing a useful tool for scientific discovery and analysis. They further offer an alternative approach for storage of large-scale atmospheric chemistry data. Importantly, randomized methods are an efficient alternative to distributed computing if these computational resources are not available. For instance, (Gittens et al., 2018) can compute the SVD of a 2.2 TB terabyte data-set in about 60 seconds, given a super computer with many nodes. However, if super computing is not available, our method offers an attractive alternative which does not require expensive compute hours on a cluster.

The paper is outlined as follows: Sec. 2 gives an overview of the global chemistry simulation engine used to produce the data of interest. Section 3 highlights the various decomposition methods that can be produced using randomized linear algebra techniques. Section 4 shows the results of the dimensionality reduction procedures, highlighting the effectiveness of each technique. Section 5 shows how such techniques can be used for data compression and reduced order models, enabling compact representations of the data for a variety of broader scientific studies. Section 6 provides concluding remarks and a brief outlook for data sciences applied to atmospheric dynamics and global chemistry analysis.

## 2  Atmospheric Chemistry Model and Data

Understanding the composition of the atmosphere is critical for a wide range of applications, including air quality, chemistry-climate interactions, and global biogeochemical cycling. Chemical transport models (CTM) are used to simulate the evolution of atmospheric constituents in space and time (Brasseur and Jacob, 2017). A CTM solves the system of coupled continuity equations for an ensemble of $m$ species with number density vector $\mathbf{n} = (n_1, \ldots, n_m)^T$ via operator splitting of transport and local processes:

$$\frac{\partial n_i}{\partial t} = -\nabla \cdot (n_i \mathbf{U}) + (P_i - L_i)(\mathbf{n}) + E_i - D_i \qquad i \in [1, m] \tag{2}$$

with $\mathbf{U}$ being the wind vector, $(P_i - L_i)(\mathbf{n})$ the (local) chemical production and loss terms, $E_i$ the emission rate, and $D_i$ the deposition rate of species $i$. The transport operator,

$$\frac{\partial n_i}{\partial t} = -\nabla \cdot (n_i \mathbf{U}) \qquad i \in [1, m] \tag{3}$$

involves spatial coupling across the model domain but no coupling between chemical species, while the chemical operator,

$$\frac{dn_i}{dt} = (P_i - L_i)(\mathbf{n}) + E_i - D_i \qquad i \in [1, m] \tag{4}$$





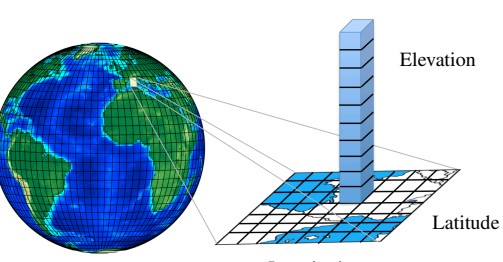

**Figure 1.** Atmospheric chemistry simulation on a global mesh with discretized longitude, latitude and elevation. Each illustrated grid cell contains time-series data for the atmospheric chemistry dynamics. Well resolved simulations generate massive data sets that are often not amenable to diagnostic analysis. Our proposed algorithms offer a scalable architecture for the analysis of global spatio-temporal data.

includes no spatial coupling but the species are chemically linked through a system of ordinary differential equations (ODEs).

Chemistry models repeatedly solve equations (3) and (4), which requires full knowledge of the chemical state of the atmosphere at all locations and times. The resulting 4-dimensional data sets (longitude,latitude,levels,species) can become massive, which makes it unpractical to output them at high temporal frequency. As a consequence, model output is generally restricted to a few selected species of interest (e.g. ozone), while the full model state is only output very infrequently, e.g. to archive the information for future model restarts ('restart file'). We show here that the chemical state of a CTM such as GEOS-Chem has distinct low-ranked features and exploiting these properties using modern diagnostic tools such as variable reduction or sub-sampling makes it possible to represent the same amount of information in a computationally more efficient manner. While we focus here on identifying low-ranked features across the spatio-temporal dimension - i.e. for each species separately – the presented methods could similarly (and independently) be applied across the species domain.

## 2.1 Global Atmospheric Chemistry Simulations

The reference simulation of atmospheric chemistry was generated using the GEOS-Chem model. GEOS-Chem (http://geos-chem.org) is an open-source global model of atmospheric chemistry that is used by over a hundred active research groups in 25 countries around the world for a wide range of applications. The code is freely available through an open license (http://acmg.seas.harvard.edu/geos/geos_licensing.html). GEOS-Chem can be run in offline mode as a chemical transport model (CTM)  (Bey, 2001; Eastham et al., 2018) or as an online component within the NASA Goddard Earth System Model (GEOS)  (Long et al., 2015; Hu et al., 2018). We use here the offline version of GEOS-Chem v11-01, driven by archives of assimilated meteorological data from the GEOS Forward Processing (GEOS-FP) data stream of the NASA Global Modeling and Assimilation Office (GMAO). The model chemistry scheme includes detailed HOx-NOx-VOC-ozone-BrOx tropospheric chemistry as originally described by Bey (2001) and with addition of BrOx chemistry by Parrella et al. (2012) and updates to isoprene oxidation as described by Mao et al. (2013). Dynamic and chemical time steps are 30 and 20 minutes, respectively. Stratospheric chemistry is modelled using a linearized mechanism as described by Murray et al. (2012).

We performed a one-year simulation of GEOS-Chem (July 2013 - June 2014) at $4° \times 5°$ horizontal resolution to generate a comprehensive set of atmospheric chemistry model diagnostics. For every chemistry time step, the concentrations of all 143 chemical constituents were archived immediately before and after chemistry in units of molecules/cm$^3$. The difference between these concentration pairs are the species tendencies due to chemistry (expressed in units of molecules/cm$^3$/s). Since the solution of chemical kinetics is also a function of the environment, we further output key environmental variables such as temperature, pressure, water vapor, and photolysis rates. The latter are computed online by GEOS-Chem using the Fast-JX code of Bian and Prather (2002) as implemented in GEOS-Chem by Mao et al. (2010) and Eastham et al. (2014). At every time step, the data set thus consists of nfeatures $= 143 + 91 + 3 + 143 = 380$ data points at every grid location. We restrict our analysis to the lowest 30 model levels to avoid influence from the stratosphere. The resulting data set has dimensions nlon $\times$ nlat $\times$ nlev $\times$ ntimes $\times$ nfeatures $= 72 \times 46 \times 30 \times 26280 \times 380 = 9.9 \times 10^{11}$.



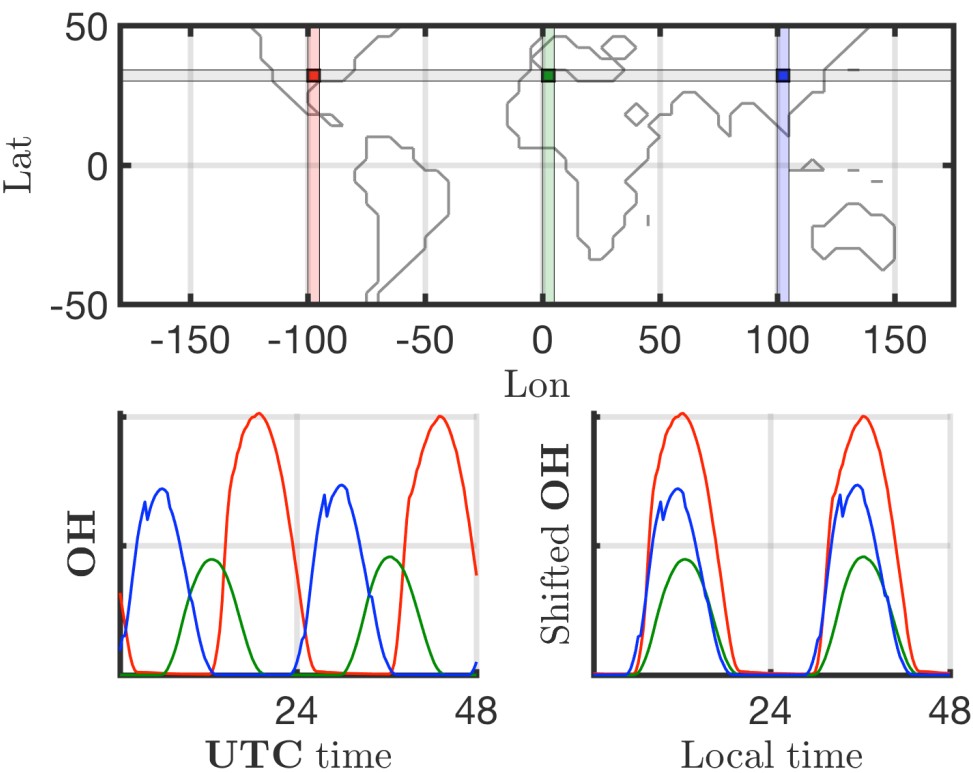

**Figure 2.** *Shifting the data for each cell in time to align the local time zones across a latitude to the prime meridian*(Lon = 0) *local time, shown here for* **OH** *absolute concentration for* Lat = 30

## 2.2 Data preprocessing

Many dimensionality reduction techniques rely on an underlying singular value decomposition of the data that extracts correlated patterns in the data. A fundamental weakness of such SVD-based approaches is the inability to efficiently handle invariances in the data (Kutz, 2013). Specifially, translational and/or rotational invariances of low-rank features in the data are
5 not well captured  (Kutz, 2013; Kutz et al., 2016). One of the key environmental variables driving the chemistry is photolysis rate, the absolute concentrations of many chemicals of interest accordingly 'turn on' and are non zero during day time, and 'turn off' or go to zero during the night. The time series of absolute chemical concentrations exhibit a translating wave traversing the globe from east to west with constant velocity. The time series for the chemical species **OH** (hydroxyl radical) is plotted with respect to UTC time for one latitude/elevation and three different longitudes on bottom left in Fig. 2, highlighting the trans-
10 lational invariance in the absolute concentration data. Any SVD- based approach will be unable to capture this translational invariance and correlate across snapshots in time, producing an artificially high dimensionality, i.e., higher number of modes would be needed to characterize the dynamics due to translation (Kutz, 2013). To overcome this issue the time series for each grid point are shifted to align with the local GMT time, as shown on bottom right in Fig. 2. With the local times for each grid point aligned SVD-based dimensionality reduction techniques can now identify and isolate coherent low-dimensional features
in the data.

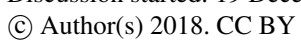 

## 3 Scalable Matrix Decompositions for Diagnostics

The following subsections detail a probabilistic framework for matrix decompositions that includes a nonnegative matrix factorization as well as a sparsity-promoting technique. The mathematical architectures proposed provide scalable computational tools for the analysis of global chemistry dynamics.

### 3.1 Probabilistic framework for low-rank approximations

Assume that the data matrix $\mathbf{X} \in \mathbb{R}^{m \times n}$ has rank $r$, where $r \leq \min\{m, n\}$. The objective of a low-rank matrix approximation to the input data matrix $\mathbf{X}$ is to find two smaller matrices

$$\underset{m \times n}{\mathbf{X}} \quad \approx \quad \underset{m \times r}{\mathbf{E}} \quad \underset{r \times n}{\mathbf{F}} \tag{5}$$

where the columns of $\mathbf{E}$ spans the column space of $\mathbf{X}$, and the rows of $\mathbf{F}$ spans the row space of $\mathbf{X}$. These factors can be stored much more efficiently, and can be used to approximate the massive input data matrix and summarize the interesting low-dimensional features which are often interpretable. Probabilistic algorithms have been established over the past two decades to compute such computationally tractable smaller matrix approximations. We seek a near-optimal low dimensional approximation of the input data matrix $\mathbf{X}$ using a probabilistic framework as formulated by (Halko et al., 2011). Conceptually, the probabilistic framework splits the task of to computing a near-optimal low rank approximation into two logical stages:

– **Stage A**: Compute a low dimensional subspace that approximates the column space of $\mathbf{X}$. We aim to find a near-optimal basis $\mathbf{Q} \in \mathbb{R}^{m \times k}$ with orthonormal columns such that

$$\mathbf{X} \approx \mathbf{Q}\mathbf{Q}^{\mathsf{T}}\mathbf{X} \tag{6}$$

is satisfied, where $k$ is the desired target rank. Random projections are used to sample the column space of the input matrix $\mathbf{X}$. Random projections are data agnostic, constructed by first drawing a set of $k$ independent random vectors $\{\boldsymbol{\omega}_i\}_{i=1}^{k}$, for instance, from the standard normal distribution; then mapping $\mathbf{X}$ to the low dimensional space to obtain the random sample projections $\mathbf{y}_i := \mathbf{X}\boldsymbol{\omega}_i$ for $i = 1, \ldots, k$. Define a random test matrix $\boldsymbol{\Omega} = [\boldsymbol{\omega}_1, \ldots, \boldsymbol{\omega}_k] \in \mathbb{R}^{n \times k}$ where the sample random projections form the sampling matrix $\mathbf{Y} \in \mathbb{R}^{m \times k}$ are given by

$$\mathbf{Y} := \mathbf{X}\boldsymbol{\Omega} \tag{7}$$

$\mathbf{Y}$ is denoted as the *sketch matrix*. The columns of $\mathbf{Y}$ are now orthonormalized using the QR-decomposition $\mathbf{Y} = \mathbf{Q}\mathbf{R}$, where $\mathbf{Q}$ is the near-optimal low dimensional basis that approximates the column space of the input data matrix. For most real-world data matrices with gradually decaying singular value spectrum, this basis matrix $\mathbf{Q}$ does not provide a good approximation for the column space of the input data matrix. A much better approximation is obtained by:

– *Oversampling*: For target rank $k$, for most data matrices we may have non-zero singular values $\{\sigma_i\}_{i=k+1}^{\min(m,n)}$. As a consequence, the sketch $\mathbf{Y}$ obtained above does not exactly span the column space of the input data matrix. Oversampling, *i.e.*, using $l = k + p$ random projections to form the sketch overcomes this issue, and a small number of additional projections $p = \{5, 10\}$ is often sufficient to obtain a good basis comparable to the best possible basis (Martinsson, 2016).

– *Power iteration scheme*: The quality of $\mathbf{Q}$ can be improved by the concept of power sampling iterations (Gu, 2014; Halko et al., 2011; Rokhlin et al., 2010). An improved sketch is defined under this concept as $\mathbf{Y} := \mathbf{X}^{(q)}\boldsymbol{\Omega}$, where $q$ is an integer specifying the number of power iterations. This process enforces a more rapid decrease of the singular values, enabling the algorithm to sample the relevant information related to the dominant singular values while the unwanted information is suppressed. As few as $q = \{1, 2, 3\}$ power iterations can considerably improve the accuracy of the approximation. Orthogonalizing the sketch between each iteration further improves the numerical stability of the algorithm.



– **Stage B**: At this stage, we form a smaller matrix $\mathbf{B}$

$$\mathbf{B} := \mathbf{Q}^\mathsf{T}\mathbf{X} \in \mathbb{R}^{l \times n} \tag{8}$$

*i.e.*, restrict the high-dimensional input matrix to the low-dimensional space spanned by the near-optimal basis $\mathbf{Q}$ obtained in Stage A. Geometrically, this is a projection which takes points in a high dimensional measurement space to a low-dimensional space while maintaining the structure in a Euclidean sense.

The probabilistic framework detailed above is referred to as the QB decomposition of the input data matrix $\mathbf{X}$, and yields the following low-rank approximation

$$\underset{m \times n}{\mathbf{X}} \quad \approx \quad \underset{m \times l}{\mathbf{Q}} \quad \underset{l \times n}{\mathbf{B}} \tag{9}$$

Note that the randomized algorithm outlined here requires two passes over the entire data matrix to construct the basis matrix $\mathbf{Q}$. The near-optimal low rank approximation $\mathbf{B} \in \mathbb{R}^{l \times n}$, where $l \ll \min(m, n)$, can now be used instead of the data matrix $\mathbf{X}$ to compute traditional deterministic matrix decompositions for data analysis. The QB decomposition can also be extended to distributed and parallel computing, see (Voronin and Martinsson, 2015). (Martinsson, 2016) provides a simplified description of the expected error:

$$\mathrm{E}\|\mathbf{X} - \mathbf{QB}\|_2 \leq \left[1 + \sqrt{\tfrac{k}{p-1}} + \tfrac{e\sqrt{k+p}}{p} \cdot \sqrt{n-k}\right]^{\frac{1}{2q+1}} \sigma_{k+1}(\mathbf{X}) \tag{10}$$

As $p$ increases, the error tends towards the best approximation error $\sigma_{k+1}(\mathbf{X})$.

### 3.2 Randomized Singular Value Decomposition

The data matrix $\mathbf{X} \in \mathbb{R}^{m \times n}$ has a singular value decomposition (SVD) of the form

$$\mathbf{X} = \mathbf{U}\boldsymbol{\Sigma}\mathbf{V}^\mathsf{T} \tag{11}$$

with unitary matrices $\mathbf{U} = [\mathbf{u}_1, \ldots, \mathbf{u}_m] \in \mathbb{R}^{m \times m}$ and $\mathbf{V} = [\mathbf{v}_1, \ldots, \mathbf{v}_n] \in \mathbb{R}^{n \times n}$ orthonormal such that $\mathbf{U}^\mathsf{T}\mathbf{U} = \mathbf{I}$ and $\mathbf{V}^\mathsf{T}\mathbf{V} = \mathbf{I}$. The left singular vectors in $\mathbf{U}$ provide a basis for the range (column space), and the right singular vectors in $\mathbf{V}$ provide a basis for the domain (row space) of the data matrix $\mathbf{X}$. The rectangular diagonal matrix $\boldsymbol{\Sigma} \in \mathbb{R}^{m \times n}$ has the corresponding non-negative singular values $\sigma_1 \geq \ldots \geq \sigma_n \geq 0$, which describe the spectrum of the data. Low-rank matrices have rank $r$ that is much smaller than the dimension of the measurement space, *i.e.*, $r \ll m, n$ and the singular values $\{\sigma_i :\geq r + 1\}$ are zero. The corresponding singular vectors span the left and right null spaces of the matrix. In practical applications the data matrix are often contaminated by errors making it's effective rank smaller than the exact rank $r$. In such cases the matrix can be well approximated by only those singular vectors which correspond to the singular values of a significant magnitude, and a reduced version of the SVD is computed

$$\mathbf{X}_k := \mathbf{U}_k\boldsymbol{\Sigma}_k\mathbf{V}_k = [\mathbf{u}_1, \ldots, \mathbf{u}_k]\,\mathrm{diag}(\sigma_1, \ldots, \sigma_k)\,[\mathbf{v}_1, \ldots, \mathbf{v}_k]^\mathsf{T} \tag{12}$$

where $k$ denotes the desired target rank of the approximation. Choosing an optimal $k$ is highly dependent on the task. If a highly accurate reconstruction of the original data is desired, then $k$ should be chosen closer to the effective rank of the data matrix. On the other hand if a very low dimensional representation of dominant features is desired, then $k$ might be chosen to be much smaller. The Eckart-Young theorem (Eckart and Young, 1936) states that the low-rank SVD provides the optimal rank-$k$ reconstruction of a matrix in the least-squares sense

$$\mathbf{X}_k := \underset{\mathrm{rank}(\mathbf{X}_k')}{\mathrm{argmin}} \left\|\mathbf{X} - \mathbf{X}_k'\right\| \tag{13}$$

with the reconstruction error in the spectral and Frobenius norm given by





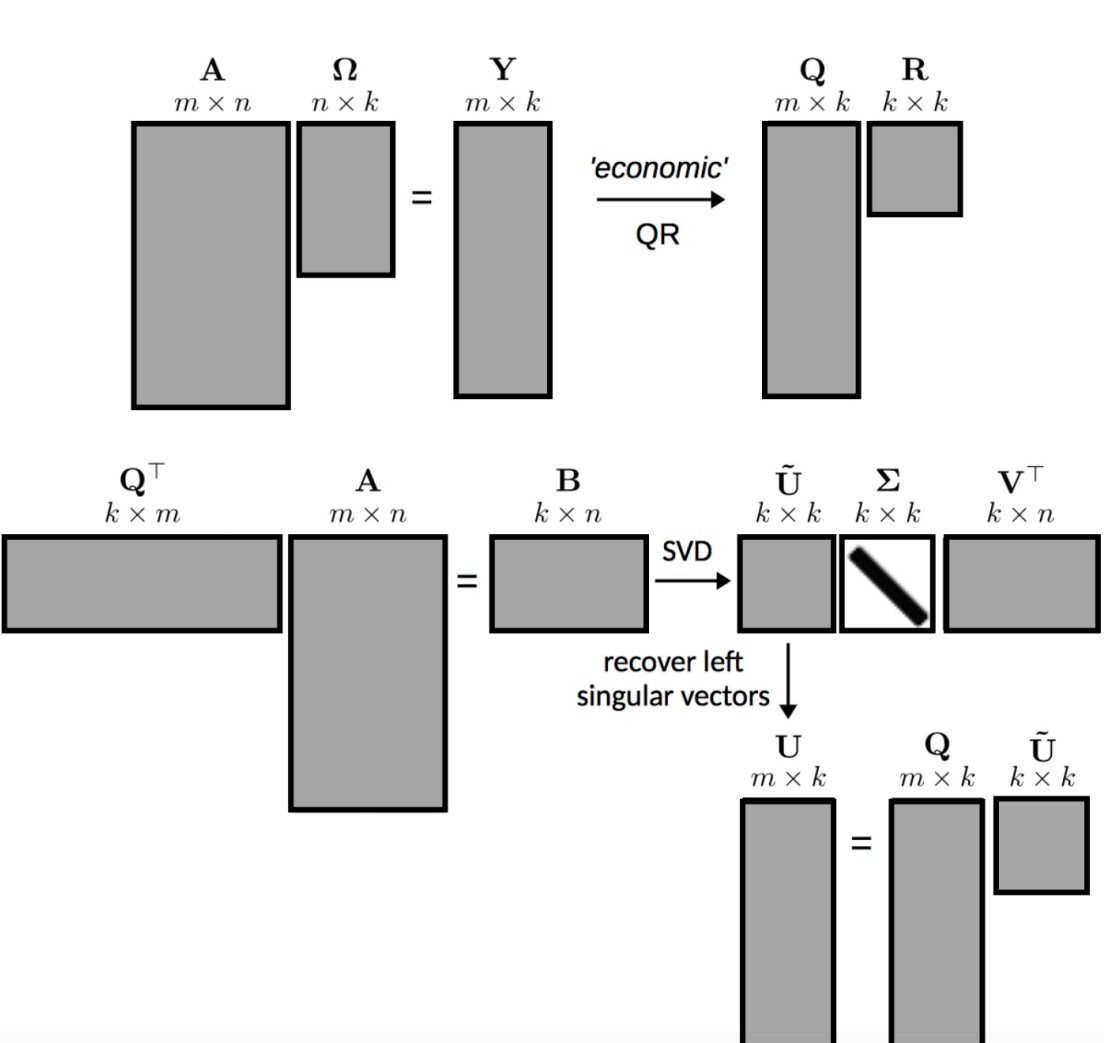

**Figure 3.** Illustration of the randomized matrix decomposition technique. The random sampling matrix $\boldsymbol{\Omega}$ is used to produce a new matrix $\mathbf{Y}$ which can be decomposed using a $QR$ decomposition. This leads to the construction of the matrix $\mathbf{B}$ which is used for approximating the left and right singular vectors.



[t]

$$\|\mathbf{X} - \mathbf{X}_k\|_2 = \sigma_{k+1}(\mathbf{X}) \text{ and } \|\mathbf{X} - \mathbf{X}_k\|_F = \sqrt{\sum_{j=k+1}^{\min(m,n)} \sigma_j^2(\mathbf{X})} \tag{14}$$

For massive datasets, however, the cost of computing the full SVD of the data matrix $\mathbf{X}$ is order $O\left(mn^2\right)$, from which the first $k$ components can then be extracted to form $\mathbf{X}_k$. Randomized algorithms are computational efficient and 'surprisingly'
reliable, these techniques can be used to obtain an approximate rank-$k$ SVD at a substantially more efficient cost of $O\left(mnk\right)$. The randomized low-rank SVD (rSVD) algorithm has favorable error bounds relative to the optimal truncated SVD. The probabilistic framework is used to obtain a near-optimal low rank approximation $\mathbf{B} \in \mathbb{R}^{l \times n}$, where $l \ll \min(m,n)$. This can now be used instead of the data matrix $\mathbf{X}$, and a full SVD of $\mathbf{B}$ is computed

$$\mathbf{B} = \tilde{\mathbf{U}} \boldsymbol{\Sigma} \mathbf{V}^{\mathsf{T}} \tag{15}$$

to give the first $l$ right singular vectors $\mathbf{V} \in \mathbb{R}^{n \times l}$ and the corresponding singular values $\boldsymbol{\Sigma} \in \mathbb{R}^{l \times l}$. The left singular vectors $\mathbf{U} \in \mathbb{R}^{m \times l}$ are recovered from the approximate left singular vectors $\tilde{\mathbf{U}} \in \mathbb{R}^{l \times l}$ by using the near-optimal basis matrix $\mathbf{Q}$

$$\mathbf{U} \approx \mathbf{Q}\tilde{\mathbf{U}} \tag{16}$$

For the absolute concentration data matrix, note that the right singular vectors $\mathbf{V}$ are temporal and the left singular vectors $\mathbf{U}$ are the spatial dominant features of the system. We also compute a cumulative energy spectrum from the singular values, the energy in the first $j$ dominant modes is given by:

$$\frac{\sum_{i=1}^{j} \sigma_i^2}{\text{Total Energy in the Data}} \tag{17}$$

where $\mathbf{S}$ is the main diagonal of the singular value matrix $\boldsymbol{\Sigma}$ and the total energy in the data is computed from the Frobenius norm as $\|\mathbf{X}\|_F^2$.

The algorithm architecture is conceptually outlined in Fig. 3. This shows the basic architecture and the structure which allows for a rapid approximation of the left and right singular values and eigenvectors.

### 3.3 Randomized Nonnegative Matrix Factorization

A significant drawback of commonly used dimensionality reduction techniques, such as SVD based Principal Component Analysis (PCA) *etc.*, is that they permit both positive and negative terms in their components. In many data applications, such as in the absolute concentration, negative terms fail to be interpretable in a physically meaningful sense, i.e. chemical concentrations are not negative. To address this problem the set of basis vectors are constrained to nonnegative terms (Lee and Seung, 1999; Paatero and Tapper, 1994), this paradigm is the nonnegative matrix factorization (NMF). NMF has emerged as a powerful dimension reduction tool that allows computation of sparse, parts-based representation of physically meaningful additive factors that describe coherent structures within the data. Given the data matrix $\mathbf{X} \in \mathbb{R}^{m \times n}$, the NMF has to find two matrices of a much lower rank

$$\underset{m \times n}{\mathbf{X}} \approx \underset{m \times k}{\mathbf{W}} \underset{k \times n}{\mathbf{H}} \tag{18}$$

where $k$ is the target rank. The SVD finds an exact solution of this problem in the least-squares sense, as detailed in the previous section, but the resulting factors are not guaranteed to be physically meaningful, i.e. positive values. NMF on the other hand gives an additive parts-based representation of the data that preserves useful properties such as sparsity and nonnegativity by imposing additional nonnegativity constraints: $\mathbf{W} \geq \mathbf{0}$ and $\mathbf{H} \geq \mathbf{0}$. The sparse parts-based features have an intuitive interpretation which have been exploited in environmental modeling (Paatero and Tapper, 1994). In environmental data, the error estimates of data can be widely varying and non-negativity is often an essential feature of the underlying models (Juntto and





Paatero, 1994; Lee et al., 1999; Paterson et al., 1999; Xie et al., 1999). Traditionally, the NMF problem is formulated as the following optimization problem:

$$
\begin{aligned}
\text{minimize} \quad & f\left(\mathbf{W}, \mathbf{H}\right) \quad = \quad \|\mathbf{X} - \mathbf{W}\mathbf{H}\|_F^2 \\
\text{subject to} \quad & \mathbf{W} \geq \mathbf{0} \quad \text{and} \quad \mathbf{H} \geq \mathbf{0}
\end{aligned}
\tag{19}
$$

This optimization problem is nonconvex and ill-posed. Since no convexification exists to simplify the optimization, no exact or unique solution is guaranteed (Gillis 2017). Different NMF algorithms, therefore, can produce distinct decompositions that minimize the objective function. Since the problem is nonconvex with respect to both factors $\mathbf{W}$ and $\mathbf{H}$, most NMF algorithms divide the problem into simpler subproblems which have closed form solutions. The convex subproblem is solved by keeping one factor fixed while updating the other, alternating and iterating until convergence. The Hierarchical Alternating Least Squares (HALS) is one variant of this method, proved to be highly efficient (Cichocki and Phan, 2009), and this is the algorithm employed here for computing the NMF.

Block coordinate descent (BCD) iterative methods fix a block of components and optimize with respect to the remaining components. The factors $\mathbf{W}$ and $\mathbf{H}$ are initialized and updated by fixing most terms except for the block comprised of the $j$th column $\mathbf{W}_{(:,j)}$ and the $j$th row $\mathbf{H}_{(j,:)}$. HALS approximately minimizes the cost function in equation (19) with respect to the remaining $k-1$ components

$$
\text{minimize } J_j\left(\mathbf{W}_{(:,j)}, \mathbf{H}_{(j,:)}\right) = \left\|\mathbf{R}^{(j)} - \mathbf{W}_{(:,j)}\mathbf{H}_{(j,:)}\right\|_F^2,
\tag{20}
$$

where $\mathbf{R}^{(j)}$ is the $j$th residual

$$
\mathbf{R}^{(j)} := \mathbf{X} - \sum_{i \neq j}^{k} \mathbf{W}_{(:,i)}\mathbf{H}_{(i,:)}
\tag{21}
$$

Gradients are derived to find the stationary points for both components, for further details we refer to (Erichson et al., 2018a).

For massive data sets randomness is again employed to replace the high-dimensional input data matrix $\mathbf{X} \in \mathbb{R}^{m \times n}$ by it's near-optimal low rank approximation $\mathbf{B} \in \mathbb{R}^{l \times n}$, where $l \ll \min(m,n)$, with the exception that the entries of $\mathbf{\Omega}$ are drawn independently from the uniform distribution in the interval $[0,1]$. We now have the following optimization problem:

$$
\begin{aligned}
\text{minimize} \quad & \tilde{f}\left(\tilde{\mathbf{W}}, \mathbf{H}\right) \quad = \quad \left\|\mathbf{B} - \tilde{\mathbf{W}}\mathbf{H}\right\|_F^2 \\
\text{subject to} \quad & \mathbf{Q}\tilde{\mathbf{W}} \geq \mathbf{0} \quad \text{and} \quad \mathbf{H} \geq \mathbf{0}
\end{aligned}
\tag{22}
$$

where the nonnegativity constraints need apply to the high dimensional factor matrix $\mathbf{W}$, but not necessarily to $\tilde{\mathbf{W}}$, since $\tilde{\mathbf{W}}$ can be rotated back to high dimensional space using the approximate relation $\mathbf{W} \approx \mathbf{Q}\tilde{\mathbf{W}}$. Since $\mathbf{Q}\mathbf{Q}^\mathsf{T} \neq \mathbf{I}$, equation (22) can only be solved approximately. The randomized HALS algorithm is formulated as

$$
\text{minimize } J_j\left(\tilde{\mathbf{W}}_{(:,j)}, \mathbf{H}_{(j,:)}\right) = \left\|\tilde{\mathbf{R}}^{(j)} - \tilde{\mathbf{W}}_{(:,j)}\mathbf{H}_{(j,:)}\right\|_F^2,
\tag{23}
$$

where $\mathbf{R}^{(j)}$ is the $j$th compressed residual

$$
\tilde{\mathbf{R}}^{(j)} := \mathbf{B} - \sum_{i \neq j}^{k} \tilde{\mathbf{W}}_{(:,i)}\mathbf{H}_{(i,:)}
\tag{24}
$$

The components are updated again by deriving the gradients. For further details, such as initialization techniques, stopping criterion and variants of HALs we refer to (Erichson et al., 2018a). For the absolute chemistry concentration data matrix, the columns of the factor $\mathbf{W}$ are the spatial modes while those of factor $\mathbf{H}$ are the temporal modes. The randomized NMF algorithm starts with an initial guess derived from a SVD of the data matrix, and returns the $\mathbf{W}$, $\mathbf{H}$ factors with columns that are not ordered. The 2-norm of the columns is computed, the columns are normalized and ordered. A product of the ordered column-wise 2-norms gives the "spectrum" for the decomposition. From this spectrum a cumulative energy spectrum is computed similar to equation (17).





### 3.4  Sparse Randomized Principal Component Analysis

Principal component analysis is a prevalent technique for dimensionality reduction, it exploits relationships among points in high-dimensional space to construct a new set of uncorrelated low-dimensional variables or principal components (PCs). The first PC explains most of the variation in the data, the second PC accounts for the second greatest variance in the data, and so on. For the data matrix $\mathbf{X} \in \mathbb{R}^{m \times n}$, which has now been centered with zero-mean, with $m$ being the number of observations and $n$ being the number of variables, the PCs $\mathbf{z}_i \in \mathbb{R}^m$ are constructed as a weighted linear combination of the original variables

$$\mathbf{z}_i := \mathbf{X}\mathbf{w}_i \tag{25}$$

where $\mathbf{w}_i \in \mathbb{R}^n$ is a vector of the corresponding weights, also denoted as modes or basis functions. Expressed concisely,

$$\mathbf{Z} := \mathbf{X}\mathbf{W} \tag{26}$$

with $\mathbf{Z} = [\mathbf{z}_1, \ldots, \mathbf{z}_n] \in \mathbb{R}^{m \times n}$ and $\mathbf{W} = [\mathbf{w}_1, \ldots, \mathbf{w}_n] \in \mathbb{R}^{n \times n}$. In most dimensionality reduction applications only the first $k$ PCs will be of interest to visualize the data in a low-dimensional space, and as the relevant features used for data clustering, classification and regression. The problem of finding the PCs can be formulated as a variance maximization problem or as a least-squares problem, *i.e.*, minimizing the sum of squared residual errors with orthogonality constraints on the weight matrix as

$$\begin{aligned} \underset{\mathbf{W}}{\text{minimize}} f(\mathbf{W}) &= \tfrac{1}{2}\left\| \mathbf{X} - \mathbf{X}\mathbf{W}\mathbf{W}^{\mathsf{T}} \right\|_{\mathrm{F}}^2 \\ \text{subject to } \mathbf{W}^{\mathsf{T}}\mathbf{W} &= \mathbf{I} \end{aligned} \tag{27}$$

The classic PCA approach outlined above generates global PCs as a linear combination of all $n$ variables, hence tends to often mix or blend various spatio-temporal scales and fails to identify and isolate underlying governing dynamics acting at each scale. Sparse principal component analysis (SPCA) is a variant which provides interpretable PCs with localized spatial support, providing a 'parsimonious' decomposition through sparsity promoting regularizers on the weights $\mathbf{W}$. Each of the sparse weight vectors $\mathbf{w}_i$ have only a few non-zero values, hence we get a linear combination of only a few of the original variables. The SPCA is mathematically formulated as a variant of PCA outlined in equation (27) as

$$\begin{aligned} \underset{\mathbf{A},\mathbf{W}}{\text{minimize}} f(\mathbf{A},\mathbf{W}) &= \tfrac{1}{2}\left\| \mathbf{X} - \mathbf{X}\mathbf{W}\mathbf{A}^{\mathsf{T}} \right\|_{\mathrm{F}}^2 + \psi(\mathbf{W}) \\ \text{subject to } \mathbf{A}^{\mathsf{T}}\mathbf{A} &= \mathbf{I} \end{aligned} \tag{28}$$

where $\mathbf{W}$ is now a sparse weight matrix and $\mathbf{A}$ is an orthonormal inverse transform matrix, i.e., the data can be approximately constructed as $\widetilde{\mathbf{X}} = \mathbf{Z}\mathbf{A}^{\mathsf{T}}$, where $\mathbf{Z}$ is the PC matrix given by equation (26). In (28), $\psi$ is a sparsity inducing regularizer such as

- $\ell_0$ norm defined as the number of non-zero elements in a vector $\mathbf{x}$, which is constrained to be $\ll n$

$$\psi_0(\mathbf{x}) = \|\mathbf{x}\|_0 \tag{29}$$

- $\ell_1$ norm, in this case the regularization problem is also known as LASSO (Least Absolute Shrinkage and Selection Operator) (Trendafilov et al., 2003)

$$\psi_1(\mathbf{x}) = \alpha \|\mathbf{x}\|_1 \tag{30}$$

where $\alpha$ controls the degree of sparsity

- The elactic net (Zou and Hastie, 2003) which is a combination of the $\ell_1$ norm and quadratic penalty

$$\psi_{\mathrm{E}}(\mathbf{x}) = \alpha \|\mathbf{x}\|_1 + \beta \|\mathbf{x}\|_2^2 \tag{31}$$

where $\alpha, \beta$ control the degree of sparsity



Note that the optimization problem in equation (28) is nonconvex and is solved similar to the NMF optimization problem by keeping one factor fixed while updating the other, alternating and iterating till convergence. For further details refer to (Erichson et al., 2018b).

For massive data sets, randomization using the probabilistic framework is employed again, where the original input data matrix $\mathbf{X}$ is projected to the range of $\mathbf{Y}$ defined in equation (7)

$$\widetilde{\mathbf{X}} = \mathbf{Q}^\mathsf{T}\mathbf{X} \tag{32}$$

reformulating equation (**??**) as

$$\underset{\mathbf{A},\mathbf{W}}{\text{minimize}}\, f\left(\mathbf{A},\mathbf{W}\right) = \rho\left(\widetilde{\mathbf{X}} - \widetilde{\mathbf{X}}\mathbf{W}\mathbf{A}^\mathsf{T}\right) + \psi\left(\mathbf{W}\right) + \varphi\left(\mathbf{A}\right) \tag{33}$$

The absolute concentration data matrix is first scaled to have mean 0. The spatial modes are the columns of matrix $\mathbf{W}$. The temporal modes or the PCs are the columns of $\mathbf{Z}$ computed from $\mathbf{X} = \mathbf{Z}\mathbf{A}^\mathsf{T}$. The minimization algortihm also formulates the problem as an eigen value problem, and returns the eigen values $\lambda_j$ associated with the $j^{th}$ mode of the decomposition, which help compute the energy spectrum of the decomposition. The energy captured by the first $j$ modes of the decomposition is computed as:

$$\frac{\sum_{i=1}^{j}\lambda_i \times (n-1)}{\text{Total Energy in the scaled Data}} \tag{34}$$

where $n$ is the total number of snapshots in time.

## 4   Data diagnostics

In this section we illustrate results from the decomposition of the GEOS-Chem model output using absolute concentration of ozone ($O_3$) as an example. The supplementary materials provides diagnostics for five additional chemicals known to dominate the global atmospheric chemistry dynamics.

Ozone is a key oxidant of the atmosphere, and high surface concentrations of $O_3$ are harmful to human health and vegetation (Avnery et al., 2011; Silva et al., 2013). Ozone production involves the photochemical oxidation of volatile organic compounds (VOCs) and carbon monoxide (CO) in the presence of nitrogen oxide radicals (NOx ≡ NO + NO₂). The chemistry of ozone is highly complex, involving hundreds of chemical species. This makes ozone a challenging compound for chemistry models (e.g. Stevenson et al., 2006; Sherwen et al., 2017; Mao et al., 2018). We find that despite the underlying complexity of the chemistry, the ozone concentration fields produced by GEOS-Chem exhibit prominent, low-ranked features.

For a given chemical species of interest the absolute concentration data matrix $\mathbf{X} \in \mathbb{R}^{m \times n}$ has dimensions $m = \text{nlon} \times \text{nlat} \times \text{nlev} = 72 \times 46 \times 30$ spatial cells, and $n = \text{number of time snapshots} = 26208$ for the year long data (one snapshot every 20 minutes).

### 4.1   Taking a logarithm of the data

For some chemical species the absolute concentration values in a small localized region dominate over the values in the rest of the grid cells. For instance, absolute concentration values of nitric oxide (NO) is several orders of magnitude higher over China and eastern Russia, as compared to those over oceans and less populated regions in the world. Correspondingly the dominant spatial modes are very localized as exhibited in the top panel of Figure 4, with only one nonzero peak over eastern Russia for the second most dominant spatial mode. SVD is unable to resolve the underlying global low order spatial features. To resolve this issue a logarithm of the data values is used instead, to bring all the concentration values to the same scale and prevent smaller signals from being damped out. The data matrix now is $\mathbf{X}_{\log} = \log(\mathbf{X} + 1)$. The second most dominant mode of the logarithm of the data as shown in the bottom panel of Figure 4 now exhibits global low order features of the data. Thus the SVD and other matrix decomposition techniques will able to identify and isolate global dominant low-order structure in the system for chemical species exhibiting localized dominant values.

### 4.2   Modes from Randomized SVD

We begin by considering the singular value spectrum and the dominant four temporal modes from the randomized SVD of the absolute concentration of ozone ($O_3$). These are presented in the top panel of Figure 5. The amount of energy explained by





the most dominant singular values gives a good indication about the low-rank nature of the underlying data. The top panel of Figure 5 shows the cumulative energy explained by the 150 most dominant singular values, as derived from randomized SVD. If all $2.7 \times 10^{11}$ model output data points were perfectly independent, each singular value would represent $1.0/2.7 \times 10^{11} = 3.7 \times 10^{-10}\%$ of the total energy. Instead, we find that the first 4 singular values combined explain 97% of the total field

energy, and the first 150 singular values capture almost 100% of the total energy. Thus, it is possible to explain 99% of the spatio-temporal structure of the highly complex ozone field with just 20 modes. These modes reveal many of the dominant features of atmospheric ozone. The bottom panel of Figure 5 illustrates the structure of the 4 dominant temporal modes. The most dominant mode (blue line) has a flat temporal structure, i.e. its importance is independent of the time of the year. The next three dominant modes all have distinct temporal patterns, i.e. they capture periodical features of atmospheric ozone. Modes 2

and 3 (red and yellow, respectively) both exhibit a frequency of 1 year, capturing features occurring on an annual basis. The 4th most dominant mode (purple) has a frequency of 6 months. Geophysical interpretation of these modes is easiest when combining the temporal pattern with the corresponding spatial features, the latter of which are shown in Figure 6. Shown are the spatial pattern of the 8 most dominant modes for the surface. It should be emphasized that the spatial patterns change with altitude, as illustrated in the supplemental material.

Surface ozone exhibits distinct seasonal patterns, which are captured by the first four modes: the first mode (top left panel in Figure 6) resembles the annual average surface concentration of ozone. It can be interpreted as time-invariant 'average ozone' field from which all other modes add or subtract to describe the spatio-temporal variability of ozone in greater detail. The second singular value (top right panel) shows a strong gradient at the equator as well as a distinct urban pattern over the Northern hemisphere (NH). The seasonal variability of this mode (peaking in August, see Figure 5) broadly follows observed

ozone burdens in the Southern hemisphere (SH) (Cooper et al., 2014), and ozone is known to increase during summertime in urban areas in the NH as a result of increased photo-chemical activity. Singular mode 3 can be seen as an additional 'forcing' to this seasonality for NH ozone: it shows dominant features over polluted areas (Europe, East China) and its seasonal amplitude complements that of singular mode 2. The most distinct feature of mode 4 is the strong pattern over Africa. We interpret this as biomass burning signal. This is supported by the frequency pattern of this mode, which shows two peaks in Jan/Feb and

Jul/Aug, which is in agreement with the two biomass burning seasons over Africa (Roberts et al., 2009).

To summarize, inspection of the spatial and temporal patterns of the dominant modes of ozone shows that randomized SVD successfully reveals prominent features of tropospheric ozone chemistry, such as elevated summertime ozone over polluted urban areas or the two biomass burning seasons over Africa. While the data set used in this study is too short to generalize the findings, these results demonstrate the potential of randomized SVD for pattern discovery of atmospheric chemistry model

output. In particular, the extent and temporal variability of the singular values can help identify highly correlated 'chemical domains' within the model, which has practical applications for model reduction considerations.

### 4.3 Modes from Randomized NMF

A drawback of the SVD solution presented in Section 4.2 is that it accepts both negative and positive solutions, which can result in physically unrealistic negative species concentrations. As discussed in Section 3.3, positive solutions can be enforced using NMF. The results from NMF of the ozone absolute concentration data are presented in Figures 7, 8. The cumulative

energy spectrum exhibited in the top panel of Figure 7 shows a much slower decay as compared to the spectrum from the SVD decomposition. This is to be expected, as NMF computes an additive parts-based representation of the low-order features in the data, which preserves sparsity in the data but requires more modes to capture the same level of energy as compared to the SVD. The four dominant temporal modes are presented in the bottom panel of Figure 7. These now capture approximately 20% of the total energy spectrum, as compared to 97% for the SVD. This is in large parts because the positivity-constraint

prevents the NMF to create a mode for annual mean ozone that can explain most of the energy spectrum - akin to mode one for SVD - but that requires both additions and subtractions from this mean field to describe ozone variations in more detail. As a result, none of the NMF modes reflects a distinct representation of the global average ozone field. This is supported by the lack of a time-invariant mode (see Figure 7) and also becomes apparent from the corresponding spatial patterns shown in Figure 8. None of those resemble the average mean ozone concentration field, as e.g. SVD mode one (see Figure 6). Still, the

first four spatial and temporal modes of NMF reflect some well known features of ozone chemistry, albeit less obvious than for

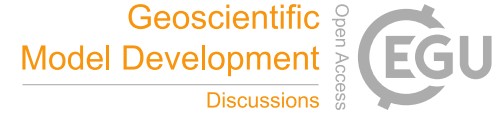

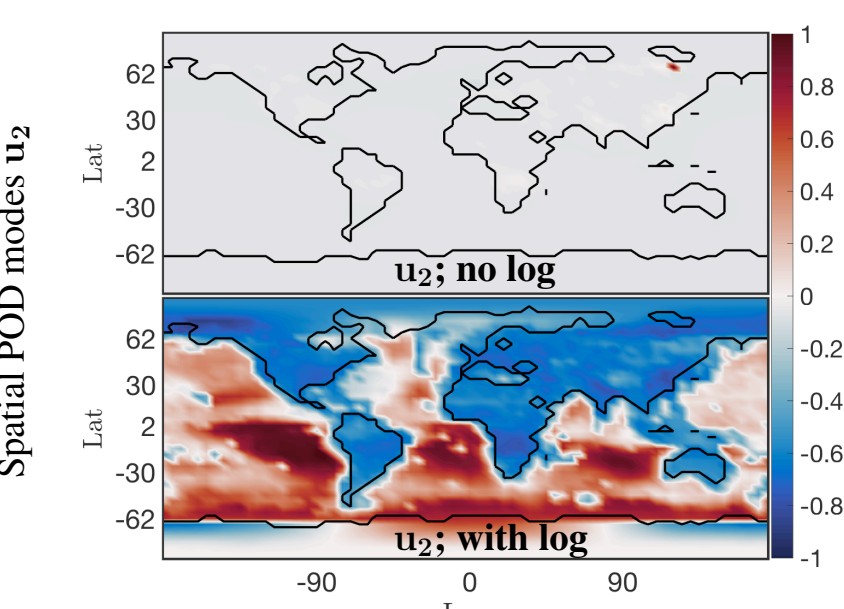

**Figure 4.** *Dominant spatial mode 2 at surface for* **NO** *absolute concentration preprocessed data before and after taking a logarithm of the preprocessed data. Taking a logarithm scales the preprocessed data so that the corresponding spatial modes exhibit the global low dimensional features, instead of only picking up on the dominant chemistry in one localized region.*

SVD. The most dominant NMF mode shows a pattern comparable to the second mode of SVD, and also has an almost identical temporal structure with a distinct peak in July/August. The second mode is almost a mirror image of the first mode, with a strong, broad-based signal in the NH that is most dominant during Mar-May but that also contributes during most other months except Jan. Mode three peaks during Sep/Oct but contributes meaningfully until February. Its spatial pattern is strongest over South America, India, Eastern China and Southern Africa, and thus captures some of the increased ozone concentrations due to fire activities (e.g. South America burning season Aug/Sep/Oct, India Oct/Nov). Mode four is similar to mode three of the SVD, with strong signals over Europe and Eastern China that peak during boreal spring.

Similar to SVD, the spatio-temporal modes of surface ozone derived from NMF reveal many of the characteristics of ozone chemistry, such as increased ozone concentrations over urban areas and biomass burning regions, as well as the seasonality of these events. Due to the strict positiveness of the solution, the signal is more muted compared to SVD, and significantly more modes are needed to reproduce the spatio-temporal pattern of ozone in detail. This makes SVD better suited for offline pattern discovery applications. However, for practical employment of reduced-order modeling techniques within an Earth System Model, we consider NMF superior since it still realistically captures ozone patterns with relatively few (10's) of modes but its concentrations are guaranteed to be positive.

### 4.4 Modes from Randomized SPCA

Spatial modes computed from the randomized SPCA are shown in Figure 10. Note the localized features isolated by SPCA in these dominant spatial modes as compared to the modes computed by the full SVD. We impose the sparsity regularizer given by



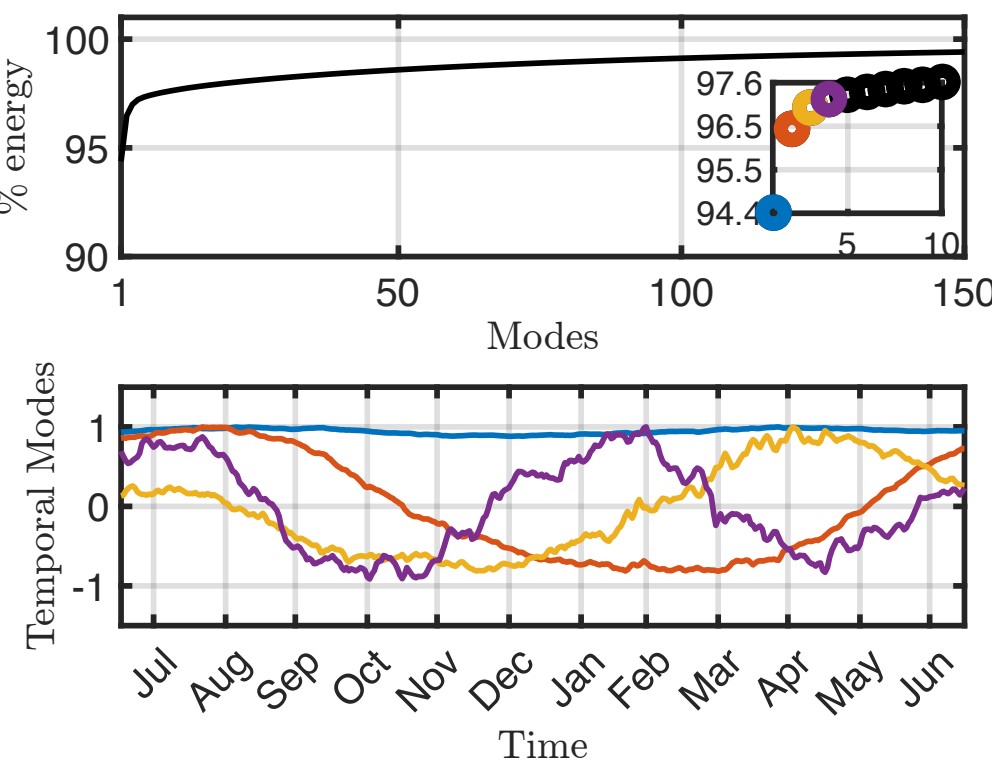

**Figure 5.** *Cumulative energy spectrum (and inset detail) of the Singular Value Decomposition (top) and the corresponding 4 dominant temporal modes (bottom) for* $O_3$ *absolute concentration preprocessed data.*

equation (31) with $\alpha = 1e-4, \beta = 1e-12$. Reducing the value of $\alpha$ gives a less sparse decomposition. The cumulative energy spectrum in the top panel of Figure 9 again demonstrates the much slower decay as compared to the SVD and more modes are needed to capture the same amount of energy due to the sparsity constraint. In terms of energy explained and interpretability of the modes, the SPCA results for ozone sit in between the results for SVD and NMF discussed above. The first four SPCA
5 modes capture more than 50% of the total energy (Figure 9), more than NMF but significantly less than SVD. As for NMF, the lower amount of energy compared to the SVD can be attributed to the fact that the SPCA does not compute a dominant mode for the mean annual ozone concentration. This is expected since SPCA is designed to capture spatially distinct features, rather than broad-based patterns. It thus 'assembles' total ozone concentrations from a series of modes that all show distinct spatial features. Of the dominant four modes shown here, the fourth one most closely resembles a generic mean concentration field
10 that contributes to the signal throughout the year (even though the signal is stronger during boreal winter). The SPCA reveals many features that are also apparent in the SVD and NMF results. The SPCA mode 1 is almost identical to mode 2 of SVD, both in spatial extent and its temporal variability. Mode 2 acts to lower ozone over Europe and Eastern China, but at a muted rate during Mar-May and also Jul/Aug. It thus has a similar effect as mode 3 of the SVD, but with opposite sign. Mode 3 can be interpreted as biomass burning signal, with its distinct hot spot over Africa and the two seasonal peaks.

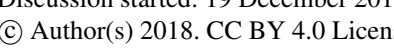


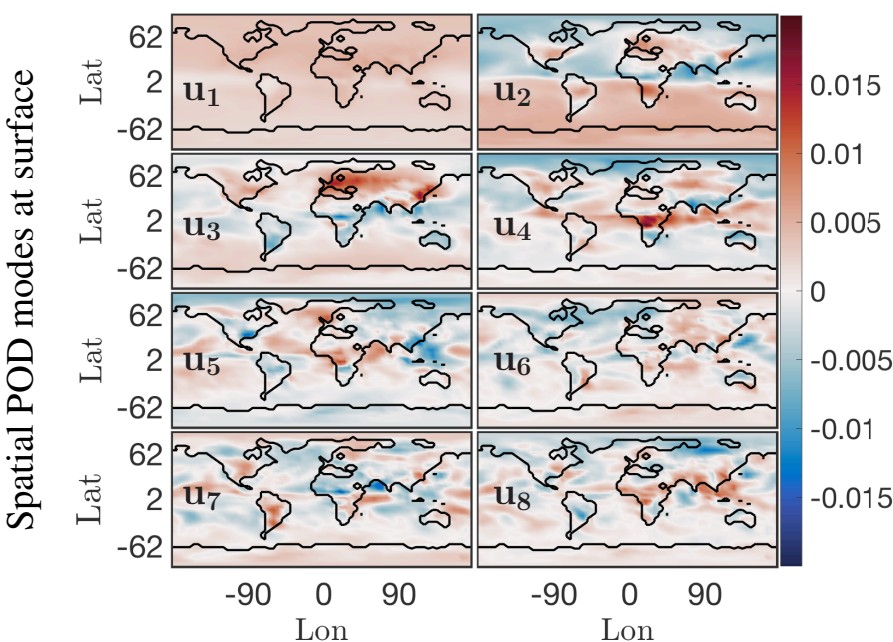

**Figure 6.** *First 8 dominant spatial modes at surface for $O_3$ absolute concentration preprocessed data. Mode 1 is the constant or mean value mode, it's corresponding temporal behavior is the blue trend in bottom panel of 5. Global low dimensional spatial features for this chemical species are exhibited in order of dominance in Modes 2 through 8.*

## 5 Data Compression and Reduced Order Modeling

Scalable diagnostic analysis is only one critically enabling aspect of the randomized decomposition methods. Indeed, the various randomized algorithms can be used to compute low-rank embeddings of the data that can be used for data compression. Thus an accurate approximation of the data can be stored at a fraction of the memory requirements of the full, high-fidelity simulation. Compression is exploited in most portable electronic formats (e.g. smart phones) by representing the data in a basis which is amenable to a sparse representation Kutz (2013). For instance, images can be massively compressed by using wavelet or Fourier basis elements since natural images are sparse in these basis elements. Compression formats such as JPEG2000 are critically enabling for the electronics industry and allowing for our electronic devices to hold an exceptionally large number of video, audio and picture files. Similarly, the scalable decomposition methods advocated here simply require a fraction of the data to be stored in the $\mathbf{Q}$ matrix and the rank-$r$ embedding columns of $\tilde{\mathbf{U}}$, $\boldsymbol{\Sigma}$ and $\mathbf{V}$. For images and video, compression allows almost perfect reconstruction of the original data while storing only a few percent of the wavelet coefficients. It is expected that similar compression performance can be achieved with the basis elements of the scalable SVD.

As an illustrative example, Figure 11(a) shows a reconstruction of the absolute concentration of surface $O_3$ at a randomly selected time using the first 5, 50 and 100 of the SVD modes, respectively, as computed from the randomized algorithm. These reconstructions require only storing 0.025%, 0.25% and 0.5% of data, respectively, as opposed to 87 million data points of the original annual surface ozone data (See Fig. 11(b)). The reconstruction with as few as 5 modes already shows that the dominant features are readily captured. It is also noted that there is virtually no difference between using 50 and 100 modes. The compression of the data with $r$ modes can be computed from the first $r$ columns of the $\mathbf{U}$ and $\mathbf{V}$ matrices along with the first $r$ diagonal terms of $\boldsymbol{\Sigma}$. This gives a data compression ratio of $(m \times n)/(m \times r + r \times n + r)$ (See Fig. 3). The compression ratio is over 4000 for 5 modes, and approximately 200 for 100 modes.



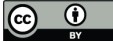

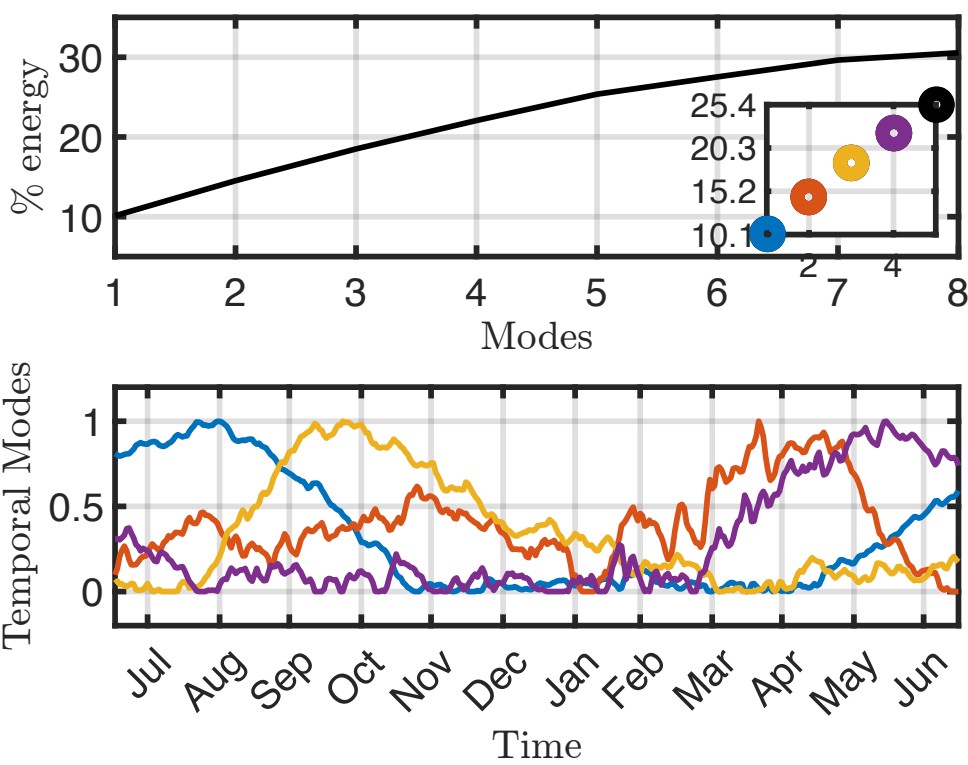

**Figure 7.** *Cumulative energy spectrum from the Nonnegative Matrix Factorization (top) and the corresponding first four columns of the ordered* **H** *temporal factor for* $O_3$ *absolute concentration preprocessed data (bottom).*

This simple example shows that the compression of modes using our randomized architecture can serve as a critically enabling tool for the storage of numerical simulations and atmospheric chemistry data, with compression rates of up to a thousand fold. This allows the real-time analysis of simulations and data sets to be performed on laptop level computing platforms. Moreover, data can be much more easily shared for collaborative purposes since file sizes can be compressed from

5   a Terabyte to only a few hundred megabytes (5 modes) to a few Gigabytes (100 modes). Such compression allows the data to be easily stored and shared on USB thumb drives.

In addition to data storage and diagnostics, the low-rank embedding spaces computed in our scalable algorithms can be used for projection-based *reduced order models* (ROMs) Benner et al. (2015). ROMs are an important emerging computational framework for solving high-fidelity, complex systems in computationally tractable ways. ROMs are especially useful for

10   enabling Monte-Carlo simulations of high-dimensional systems that have stochastic variability, such as turbulent flows. The ROMs enable computation of statistical quantities like lift and drag in turbulent flows at fraction of the computational cost. Indeed, Monte-Carlo computations of many high-dimensional problems of interest are currently intractable even with super computers, thus highlighting the need for proxy models that can be computed at reduced cost. In future work, we will aim to develop ROMs that exploit the low-rank embeddings computed with our scalable algorithms.



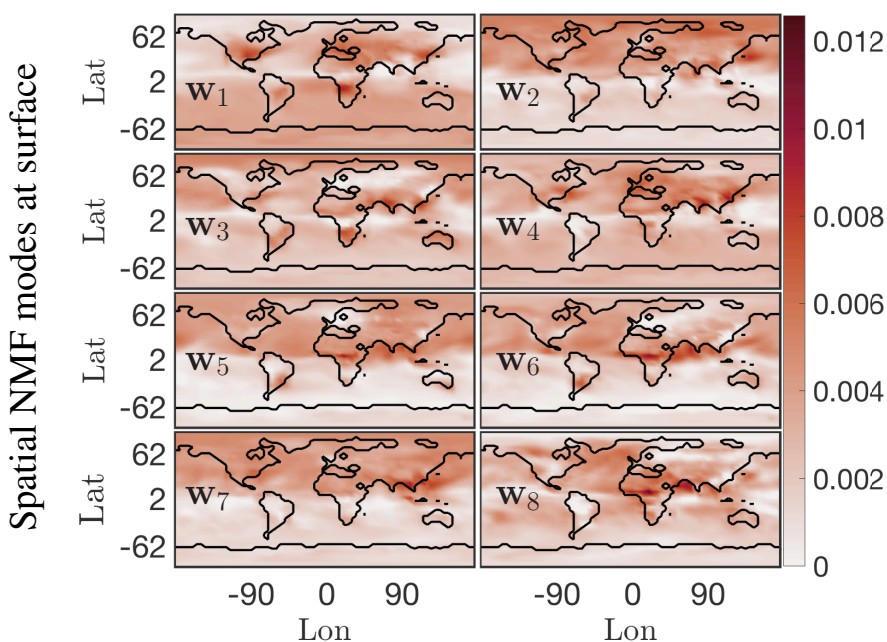

**Figure 8.** *First 8 columns of ordered* **W** *spatial factor from NMF at surface for* $O_3$ *absolute concentration preprocessed data. These modes lend themselves to easy interpretation, the most dominant mode* $w_1$ *indicates that* $O_3$ *absolute concentration is most active near eastern coastal urban China, North America and western coastal African continent around the region of Congo.*

## 6 Conclusions

Global environmental monitoring is becoming realizable through modern sensor technologies and emerging diagnostic algorithms. Despite tremendous advances and innovations, the data collection process can quickly produce volumes of data that cannot be analyzed and diagnosed in real-time, especially for applications like global atmospheric chemistry modeling which must integrate knowledge of hundreds of chemical species across a global longitude, latitude and elevation grid. This emerging *big data* era requires diagnostic tools that can scale to meet the rapidly increasing information acquired from new monitoring technologies which are producing more fine scale spatial and temporal measurements. We demonstrate a new set of diagnostic tools that are capable of extracting the dominant global features of global atmospheric chemistry dynamics. Not only are the methods scalable for both current and future sensor networks, they also have critical innovations allowing for improved interpretability, feature extraction, and data compression.

As demonstrated in this manuscript, emerging *randomized linear algebra* algorithms are critically enabling for scalable *big data* applications. The randomized algorithms exploit the fact that the data itself has low-rank features. Indeed, the method scales with the intrinsic rank of the dynamics rather than the dimension of the measurements/sensor space. Analysis of global atmospheric chemistry data shows that low-rank features indeed dominate the data. Thus full spatial mode structures can be extracted (longitude, latitude and elevation). This is in contrast to standard PCA reductions which do not scale well with the data size so that one is forced, due to computational constraints, to only analyze the data at fixed spatial features, such as looking at only a certain elevation. Alternatively, one can think of the scalable methods as being critically enabling for producing real-time analysis of emerging, streaming *big data* sets from the atmospheric chemistry community. Moreover, the dominant features of the data can be used for an efficient compression of the data for storage or reduced order modeling applications.

An important aspect of this work is that simulation data, through the GEOS-Chem model, can be used to approximate the dominant global patterns of spatio-temporal activity for individual chemicals, a collection of chemicals, or the entire chemical space. The spatio-temporal features extracted give new possibilities for understanding the interaction dynamics and





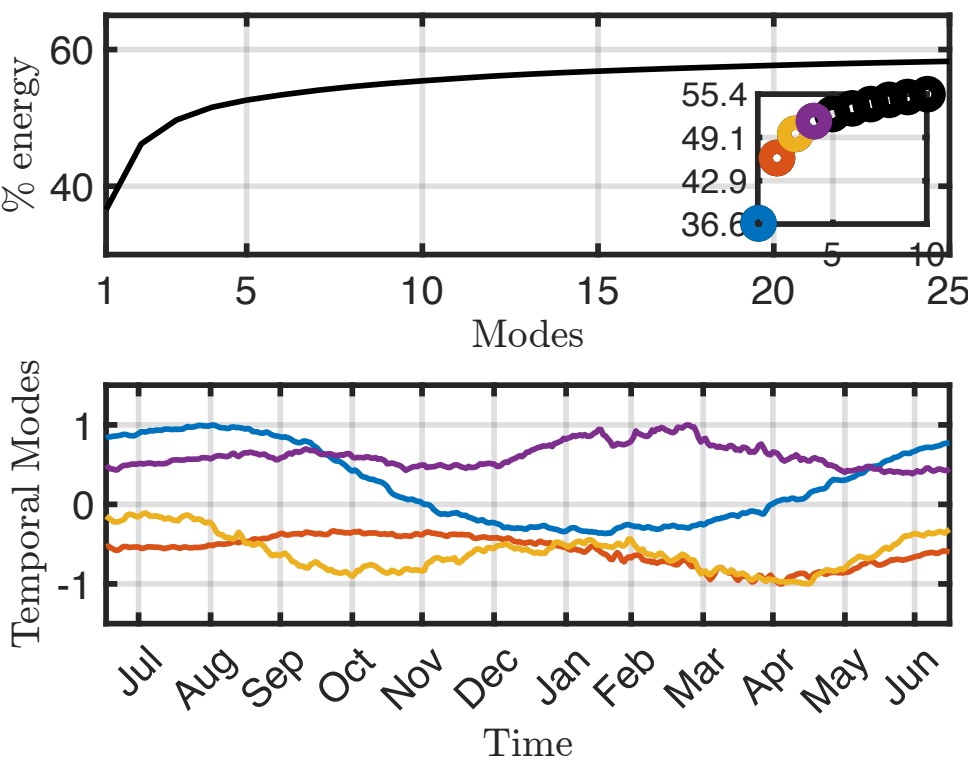

**Figure 9.** *Cumulative energy spectrum from the Sparse Principal Component Anaylsis (top) and the corresponding 4 dominant temporal modes (bottom) for* $O_3$ *absolute concentration preprocessed data.*

relevant spatial regions where various chemical dynamics are important. This gives new possibilities for scientific discovery and understanding of the complex processes driving the global chemistry profile.

*Code and data availability.* The code to run and generate the results presented in the manuscript, plot all the figures in the paper, and produce the three randomized matrix decompositions for Ozone is linked to the github account of Meghana Velager:

https://github.com/mvelegar/ScalableDiagnostics

The randomized algorithms used can be found on the github account of Benjamin Erichson:

10   https://github.com/erichson/ristretto

An example data set of the unprocessed simulation data (one month of Ozone dynamics), can be downloaded here:

https://www.dropbox.com/sh/jmx8j0d9ep72eo/4/AACvHijg_i



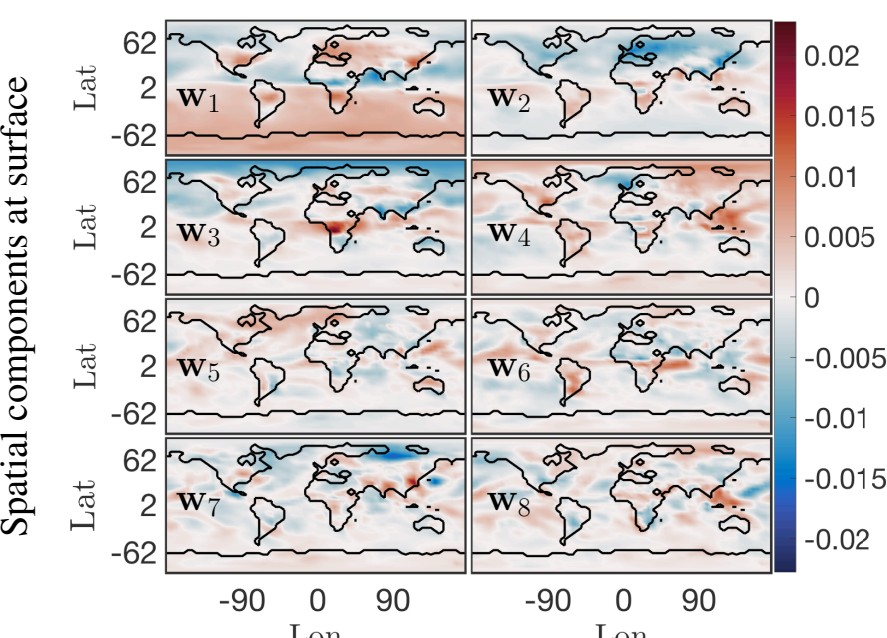

**Figure 10.** *First 8 principal components from SPCA at surface for* $O_3$ *absolute concentration preprocessed data. With the sparsity constraint these spatial modes exhibit only localized low dimensional features as compared to those from the SVD of the data. Compare the SVD mean value Mode 1* $\mathbf{u}_1$ *from 6 which exhibits a more or less constant field as the dominant low dimensional global feature, to SPCA Mode 1* $\mathbf{w}_1$ *here which picks up on localized dominant features in the data. The corresponding temporal SPCA mode 1 also exhibits a seasonal variation.*

*Author contributions.* MV, CK and JNK designed the numerical and data extraction experiment, and MV and BE carried them out. CK developed the geos-CHEM model code and performed the simulations for generating the data, while MV and BE designed the randomized algorithms and pre-processing steps required for the analysis. All authors contributed to the preparation of the manuscript.

*Competing interests.* The authors have no competing interests for this work.

*Acknowledgements.* J. N. Kutz acknowledges support from the Air Force Office of Scientific Research (AFOSR FA9550-17-1-0329). C. A. ₅
Keller acknowledges support by the NASA Modeling, Analysis, and Prediction (MAP) Program.

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





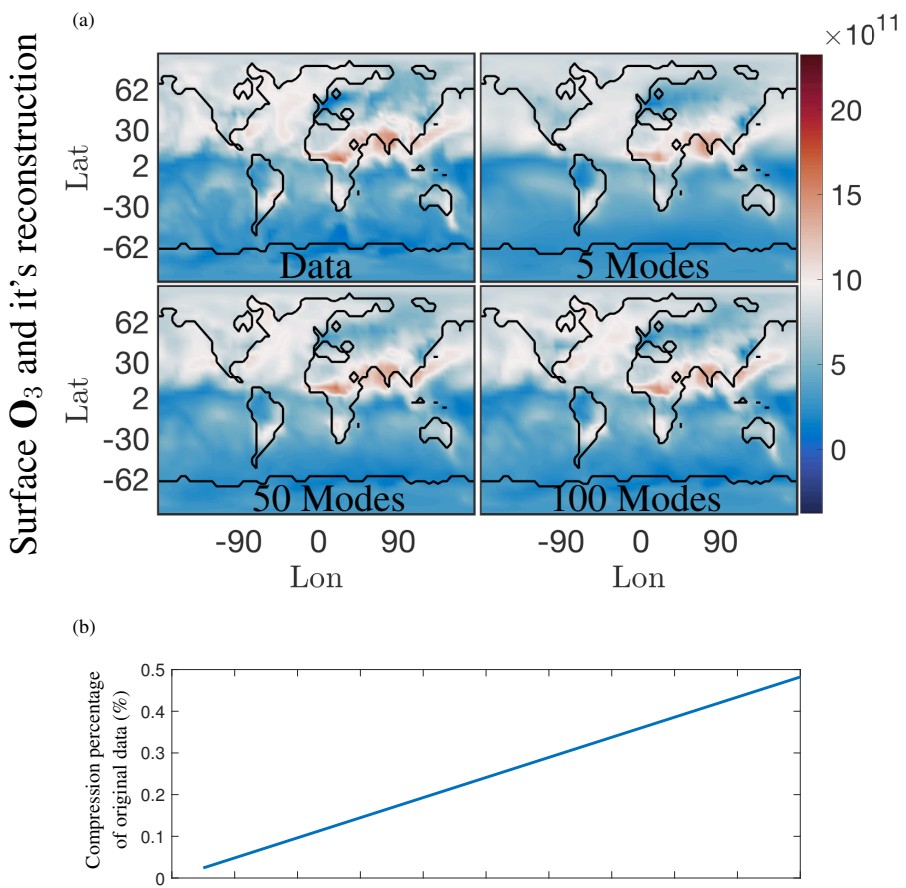

**Figure 11.** (a) *One-time snapshot of surface* $O_3$ *absolute concentration reference data (top left) and its reconstruction using* **5, 50** *and* **100** *SVD modes, respectively. Using 5 modes, only the most dominant features are reconstructed successfully, but as the number of modes used for reconstruction increases more of the finer local features in the original data are picked up. Similar results hold for both SPCA and NMF. (b) Compression percentage of the original data (%) as a function of the rank of the modes retained. For the **5, 50** and **100** modes illustrated in (a), the data can be compressed into as little as 0.025% for five modes, and 0.5% for 100 modes.*

Bian, H. and Prather, M. J.: Fast-J2: Accurate Simulation of Stratospheric Photolysis in Global Chemical Models, Journal of Atmospheric Chemistry, 41, 281–296, https://doi.org/10.1023/A:1014980619462, https://doi.org/10.1023/A:1014980619462, 2002.

Brasseur, G. P. and Jacob, D. J.: Modeling of Atmospheric Chemistry, Cambridge University Press, 2017.

Cichocki, A. and Phan, A. H.: Fast Local Algorithms for Large Scale Nonnegative Matrix and Tensor Factorizations., IEICE Transactions,
5    92-A, 708–721, http://dblp.uni-trier.de/db/journals/ieicet/ieicet92a.html#CichockiP09, 2009.

Cooper, M., Martin, R. V., Wespes, C., Coheur, P.-F., Clerbaux, C., and Murray, L. T.: Tropospheric nitric acid columns from the IASI satellite instrument interpreted with a chemical transport model: Implications for parameterizations of nitric oxide production by lightning, Journal of Geophysical Research: Atmospheres, 119, 10 068–10 079, https://doi.org/10.1002/2014JD021907, http://dx.doi.org/10.1002/2014JD021907, 2014JD021907, 2014.

10  Cunningham, J. P. and Ghahramani, Z.: Linear dimensionality reduction: survey, insights, and generalizations., Journal of Machine Learning Research, 16, 2859–2900, 2015.

Drineas, P. and Mahoney, M. W.: RandNLA: randomized numerical linear algebra, Communications of the ACM, 59, 80–90, 2016.



Eastham, S. D., Weisenstein, D. K., and Barrett, S. R.: Development and evaluation of the unified tropospheric–stratospheric chemistry extension (UCX) for the global chemistry-transport model GEOS-Chem, Atmospheric Environment, 89, 52 – 63, https://doi.org/https://doi.org/10.1016/j.atmosenv.2014.02.001, http://www.sciencedirect.com/science/article/pii/S1352231014000971, 2014.

Eastham, S. D., Long, M. S., Keller, C. A., Lundgren, E., Yantosca, R. M., Zhuang, J., Li, C., Lee, C. J., Yannetti, M., Auer, B. M., Clune, T. L., Kouatchou, J., Putman, W. M., Thompson, M. A., Trayanov, A. L., Molod, A. M., Martin, R. V., and Jacob, D. J.: GEOS-Chem High Performance (GCHP): A next-generation implementation of the GEOS-Chem chemical transport model for massively parallel applications, Geoscientific Model Development Discussions, 2018, 1–18, https://doi.org/10.5194/gmd-2018-55, https://www.geosci-model-dev-discuss.net/gmd-2018-55/, 2018.

Eckart, C. and Young, G.: The approximation of one matrix by another of lower rank, Psychometrika, 1, 211–218, https://doi.org/10.1007/BF02288367, 1936.

Erichson, N. B., Voronin, S., Brunton, S. L., and Kutz, J. N.: Randomized matrix decompositions using R, arXiv preprint arXiv:1608.02148, 2016.

Erichson, N. B., Brunton, S. L., and Kutz, J. N.: Compressed singular value decomposition for image and video processing, in: Computer Vision Workshop (ICCVW), 2017 IEEE International Conference on, pp. 1880–1888, IEEE, 2017.

Erichson, N. B., Mendible, A., Wihlborn, S., and Kutz, J. N.: Randomized nonnegative matrix factorization, Pattern Recognition Letters, 104, 1–7, 2018a.

Erichson, N. B., Zeng, P., Manohar, K., Brunton, S. L., Kutz, J. N., and Aravkin, A. Y.: Sparse Principal Component Analysis via Variable Projection, preprint arXiv:1804.00341, 2018b.

Gittens, A., Rothauge, K., Wang, S., Mahoney, M. W., Gerhardt, L., Kottalam, J., Ringenburg, M., Maschhoff, K., et al.: Accelerating Large-Scale Data Analysis by Offloading to High-Performance Computing Libraries using Alchemist, arXiv preprint arXiv:1805.11800, 2018.

Gu, M.: Subspace Iteration Randomization and Singular Value Problems, ArXiv e-prints, 2014.

Halko, N., Martinsson, P.-G., and Tropp, J. A.: Finding structure with randomness: Probabilistic algorithms for constructing approximate matrix decompositions, SIAM review, 53, 217–288, 2011.

Hu, L., Keller, C. A., Long, M. S., Sherwen, T., Auer, B., Da Silva, A., Nielsen, J. E., Pawson, S., Thompson, M. A., Trayanov, A. L., Travis, K. R., Grange, S. K., Evans, M. J., and Jacob, D. J.: Global simulation of tropospheric chemistry at 12.5 km resolution: performance and evaluation of the GEOS-Chem chemical module (v10-1) within the NASA GEOS Earth System Model (GEOS-5 ESM), Geoscientific Model Development Discussions, 2018, 1–32, https://doi.org/10.5194/gmd-2018-111, https://www.geosci-model-dev-discuss.net/gmd-2018-111/, 2018.

Juntto, S. and Paatero, P.: Analysis of daily precipitation data by positive matrix factorization, Environmetrics, 5, 127–144, 1994.

Kutz, J. N.: Data-driven modeling & scientific computation: methods for complex systems & big data, Oxford University Press, 2013.

Kutz, J. N., Brunton, S. L., Brunton, B. W., and Proctor, J. L.: Dynamic Mode Decomposition: Data-Driven Modeling of Complex Systems, SIAM-Society for Industrial and Applied Mathematics, USA, 2016.

Lee, D. D. and Seung, S. H.: Learning the parts of objects by non-negative matrix factorization, Nature, 401, 788–791, 1999.

Lee, E., Chan, C. K., and Paatero, P.: Application of positive matrix factorization in source apportionment of particulate pollutants in Hong Kong, Atmospheric Environment, 33, 3201–3212, 1999.

Long, M. S., Yantosca, R., Nielsen, J. E., Keller, C. A., da Silva, A., Sulprizio, M. P., Pawson, S., and Jacob, D. J.: Development of a grid-independent GEOS-Chem chemical transport model (v9-02) as an atmospheric chemistry module for Earth system models, Geoscientific Model Development, 8, 595–602, https://doi.org/10.5194/gmd-8-595-2015, https://www.geosci-model-dev.net/8/595/2015/, 2015.

Mahoney, M. W. et al.: Randomized algorithms for matrices and data, Foundations and Trends® in Machine Learning, 3, 123–224, 2011.

Mao, J., Jacob, D. J., Evans, M. J., Olson, J. R., Ren, X., Brune, W. H., Clair, J. M. S., Crounse, J. D., Spencer, K. M., Beaver, M. R., Wennberg, P. O., Cubison, M. J., Jimenez, J. L., Fried, A., Weibring, P., Walega, J. G., Hall, S. R., Weinheimer, A. J., Cohen, R. C., Chen, G., Crawford, J. H., McNaughton, C., Clarke, A. D., Jaeglé, L., Fisher, J. A., Yantosca, R. M., Le Sager, P., and Carouge, C.: Chemistry of hydrogen oxide radicals ($HO_x$) in the Arctic troposphere in spring, Atmospheric Chemistry and Physics, 10, 5823–5838, https://doi.org/10.5194/acp-10-5823-2010, https://www.atmos-chem-phys.net/10/5823/2010/, 2010.

Mao, J., Paulot, F., Jacob, D. J., Cohen, R. C., Crounse, J. D., Wennberg, P. O., Keller, C. A., Hudman, R. C., Barkley, M. P., and Horowitz, L. W.: Ozone and organic nitrates over the eastern United States: Sensitivity to isoprene chemistry, Journal of Geophysical Research: Atmospheres, 118, 11,256–11,268, https://doi.org/10.1002/jgrd.50817, http://dx.doi.org/10.1002/jgrd.50817, 2013JD020231, 2013.

Mao, J., Carlton, A., Cohen, R. C., Brune, W. H., Brown, S. S., Wolfe, G. M., Jimenez, J. L., Pye, H. O. T., Lee Ng, N., Xu, L., McNeill, V. F., Tsigaridis, K., McDonald, B. C., Warneke, C., Guenther, A., Alvarado, M. J., de Gouw, J., Mickley, L. J., Leibensperger, E. M., Mathur, R., Nolte, C. G., Portmann, R. W., Unger, N., Tosca, M., and Horowitz, L. W.: Southeast Atmosphere Studies: learning from model-observation syntheses, Atmospheric Chemistry and Physics, 18, 2615–2651, https://doi.org/10.5194/acp-18-2615-2018, https://www.atmos-chem-phys.net/18/2615/2018/, 2018.

Martinsson, P.-G.: Randomized methods for matrix computations, ArXiv e-prints, 2016.

Murray, L. T., Jacob, D. J., Logan, J. A., Hudman, R. C., and Koshak, W. J.: Optimized regional and interannual variability of lightning in a global chemical transport model constrained by LIS/OTD satellite data, Journal of Geophysical Research: Atmospheres, 117, n/a–n/a, https://doi.org/10.1029/2012JD017934, http://dx.doi.org/10.1029/2012JD017934, d20307, 2012.



Paatero, P. and Tapper, U.: Positive matrix factorization: A non-negative factor model with optimal utilization of error estimates of data values, Environmetrics, 5, 111–126, https://doi.org/http://dx.doi.org/10.1002/env.3170050203, http://dx.doi.org/10.1002/env.3170050203, 1994.

Parrella, J. P., Jacob, D. J., Liang, Q., Zhang, Y., Mickley, L. J., Miller, B., Evans, M. J., Yang, X., Pyle, J. A., Theys, N., and Van Roozendael, M.: Tropospheric bromine chemistry: implications for present and pre-industrial ozone and mercury, Atmospheric Chemistry and Physics,
12, 6723–6740, https://doi.org/10.5194/acp-12-6723-2012, https://www.atmos-chem-phys.net/12/6723/2012/, 2012.

Paterson, K. G., Sagady, J. L., Hooper, D. L., Bertman, S. B., Carroll, M. A., and Shepson, P. B.: Analysis of air quality data using positive matrix factorization, Environmental Science & Technology, 33, 635–641, 1999.

Roberts, G., Wooster, M. J., and Lagoudakis, E.: Annual and diurnal african biomass burning temporal dynamics, Biogeosciences, 6, 849–866, https://doi.org/10.5194/bg-6-849-2009, https://www.biogeosciences.net/6/849/2009/, 2009.

Rokhlin, V., Szlam, A., and Tygert, M.: A Randomized Algorithm for Principal Component Analysis, SIAM Journal on Matrix Analysis and Applications, 31, 1100–1124, https://doi.org/10.1137/080736417, https://doi.org/10.1137/080736417, 2010.

Sherwen, T., Evans, M. J., Sommariva, R., Hollis, L. D. J., Ball, S. M., Monks, P. S., Reed, C., Carpenter, L. J., Lee, J. D., Forster, G., Bandy, B., Reeves, C. E., and Bloss, W. J.: Effects of halogens on European air-quality, Faraday Discuss., 200, 75–100, https://doi.org/10.1039/C7FD00026J, http://dx.doi.org/10.1039/C7FD00026J, 2017.

Silva, R. A., West, J. J., Zhang, Y., Anenberg, S. C., Lamarque, J.-F., Shindell, D. T., Collins, W. J., Dalsoren, S., Faluvegi, G., Folberth, G., Horowitz, L. W., Nagashima, T., Naik, V., Rumbold, S., Skeie, R., Sudo, K., Takemura, T., Bergmann, D., Cameron-Smith, P., Cionni, I., Doherty, R. M., Eyring, V., Josse, B., MacKenzie, I. A., Plummer, D., Righi, M., Stevenson, D. S., Strode, S., Szopa, S., and Zeng, G.: Global premature mortality due to anthropogenic outdoor air pollution and the contribution of past climate change, Environmental Research Letters, 8, 034 005, http://stacks.iop.org/1748-9326/8/i=3/a=034005, 2013.

Stevenson, D. S., Dentener, F. J., Schultz, M. G., Ellingsen, K., van Noije, T. P. C., Wild, O., Zeng, G., Amann, M., Atherton, C. S., Bell, N., Bergmann, D. J., Bey, I., Butler, T., Cofala, J., Collins, W. J., Derwent, R. G., Doherty, R. M., Drevet, J., Eskes, H. J., Fiore, A. M., Gauss, M., Hauglustaine, D. A., Horowitz, L. W., Isaksen, I. S. A., Krol, M. C., Lamarque, J.-F., Lawrence, M. G., Montanaro, V., Müller, J.-F., Pitari, G., Prather, M. J., Pyle, J. A., Rast, S., Rodriguez, J. M., Sanderson, M. G., Savage, N. H., Shindell, D. T., Strahan, S. E., Sudo, K., and Szopa, S.: Multimodel ensemble simulations of present-day and near-future tropospheric ozone, Journal
of Geophysical Research: Atmospheres, 111, https://doi.org/10.1029/2005JD006338, https://agupubs.onlinelibrary.wiley.com/doi/abs/10.1029/2005JD006338, 2006.

Trendafilov, N., Jolliffe, I. T., and Uddin, M.: A modified principal component technique based on the LASSO, Journal of Computational and Graphical Statistics, 12, 531—547, http://stats-www.open.ac.uk/staff/nt1.html, 2003.

Voronin, S. and Martinsson, P.-G.: RSVDPACK: Subroutines for computing partial singular value decompositions via randomized sampling
on single core, multi core, and GPU architectures, 2015.

Xie, Y.-L., Hopke, P. K., Paatero, P., Barrie, L. A., and Li, S.-M.: Identification of Source Nature and Seasonal Variations of Arctic Aerosol bypositive matrix factorization, Journal of the Atmospheric Sciences, 56, 249–260, 1999.

Zou, H. and Hastie, T.: Regularization and Variable Selection via the Elastic Net, Journal of the Royal Statistical Society: Series B (Statistical Methodology), 67, 301–320, 2003.