# Peer review of "Scalable Diagnostics for Global Atmospheric Chemistry using Ristretto Library (version 1.0)"

_Geoscientific Model Development, 2018_

## Referee Comment (RC1) · Anonymous Referee #1 · 8 Mar 2019

The research article entitled "Scalable Diagnostics and Data Compression for Global Atmospheric Chemistry using Ristretto Library" for global environment monitoring deals with huge volume of data. It is necessary to develop an efficient method to reduce the data obtained from the sensor for reasonable analysis. Here are some of the clarifications/minor revision required from the authors. 1. Authors have applied the already existing methods to the Global Atmospheric Chemistry Data. What is the novelty in this work compared to the existing methods?

2. What is the reason for choosing Ristretto Library and it would be better to mention the advantages of using Ristretto Library package.

3. The Comparative results with the existing techniques are necessary to prove the efficiency of the proposed method

4. Add some more related works developed in the recent years in the introduction section.

5. Any pictorial representation or block diagram need to clearly figure out your whole work.

---

## Referee Comment (RC2) · Anonymous Referee #2 · 8 Mar 2019

The paper entitled "Scalable Diagnostics and Data Compression for Global Atmospheric Chemistry using Ristretto Library" is well written. it is one of the needed research areas of current scenario. The authors are focusing to apply the dimensionality reduction for the probable compression of spatial features of the Global Atmospheric Chemistry data and for further analytics. The authors are requested to give the clarifications to the following queries for the possible acceptance of the paper.

1. The compression is achieved by means of dimensionality reduction. To achieve additional compression performance (Even the lossless entropy encoding) may be included in this proposed work to yield more compression ratio, this will even reduce the

data size.

2. "Taking logarithm of the data" is mentioned in the paper, it would be better to give detail about this along with necessary references.

3. What is the reason for representing the Global Atmospheric Chemistry Data in 2D array in this proposed work? Why didn't considered as high dimensional data such as Tensor? Please justify the reasons.

4. Mention, Among RSVD, NMF and SPCA, which is appropriate for your Global Atmospheric Chemistry Data analytics.

5. You have specified ozone (O3) in your manuscript, is there any other key species are found in the atmosphere? If so, mention it in your manuscript.

6. It would be better to provide the Bit Rate of compressed data as well as original data in a tabular form.

---

## Author Comment (AC1) · 27 Mar 2019

Referee 1 comments:

The research article entitled "Scalable Diagnostics and Data Compression for Global Atmospheric Chemistry using Ristretto Library" for global environment monitoring deals with huge volume of data. It is necessary to develop an efficient method to reduce the data obtained from the sensor for reasonable analysis. Here are some of the clarifications/minor revision required from the authors.

Thank you for the comments about the paper and for the clarifications/minor revisions

suggested.

1. Authors have applied the already existing methods to the Global Atmospheric Chemistry Data. What is the novelty in this work compared to the existing methods?

This work is aimed at bringing emerging data methods, such as the proposed randomized algorithms posited, to the atmospheric chemistry community. Much like the application of neural networks (NNs), for instance, for forecasting and prediction, the goal is to use methods developed in the computational/statistical/computer science community to large-scale scientific problems. In emerging NNs, there is typically very little innovation on the NNs themselves, but rather on what the NNs can reveal for scientific discovery. Indeed, we have demonstrated that critical diagnostics, such as the production of EOF/PCA patterns of global chemical variability, can be extracted from global data sets on laptop/desktop architectures. Unless one had access to large scale computing and HPC coding, this simply is not tractable. Our innovation is in developing a suite of methods whereby practitioners from the atmospheric sciences community can now do practical computations from desktop/laptop architectures using simple open source python code. We know of no other existing computational methods that can yield such diagnostic features on such large scale data on laptop/desktop computing.

2. What is the reason for choosing Ristretto Library and it would be better to mention the advantages of using Ristretto Library package.

The Ristretto library was developed to provide a robust and stable code base for producing a variety of dimensionality reduction algorithms from large scale data that has traditionally been intractable. Not only can it produce PCA/EOF/POD/SVD modes, but it can also produce (as demonstrated) nonnegative matrix factorization and sparse PCA versions as well. There is no other package that integrates all these features in one package. Scikit-learn does have a randomized PCA routine, but not the other physically important variants demonstrated here. Moreover, the sparse PCA routine in Scikit-learn does not perform well on the data considered here as it was developed with an eye towards other applications. As such, we instead recommend the integrated and robust package which is Ristretto.

3. The Comparative results with the existing techniques are necessary to prove the efficiency of the proposed method

We agree with this assessment. However, there currently are no techniques being used for such massive downsampling of data for porting to laptop/desktop architecture. Indeed, the current methods being used are the standard PCA/EOF/POD architecture which are computationally intractable unless used on HPC architectures. More succinctly: we have not found evidence in this field of application of other techniques (for comparison with randomized algorithms) being used for massively downsampling the data for producing dominant correlated features of the data. As for a comparison and evaluation of randomized methods, this has already been done and cited in the current work (See the citation to Erichson et al 2016).

As for distributed computing and HPC computing architectures, this is exactly what we are trying to avoid. Not only is one required to learn how to parse the computations to an HPC architecture, but few practitioners have access to such computing. Moreover, why would one do this when one can simply downsample through the randomized algorithm to achieve the desired results on laptop/desktop computing using standard python code. This is fundamentally a desktop/laptop enabled code by design. For data up to 10Tb, the data can be easily read into fast memory for producing the diagnostic features. Data that far exceeds 10Tb may require HPC architectures, but this is a significantly different world of computation than what is targeted here. Regardless, even for such enormous data sets, HPC computations can be greatly enhanced with the randomized algorithms.

4. Add some more related works developed in the recent years in the introduction section.

We have added a couple of references to very recent randomized tensor decomposi-

СЗ

tions (see response to reviewer 2, point 3). There are several directions that people have taken randomized algorithms, but they feel very far afield and perhaps inappropriate for the current application in global chemistry. But the randomized tensor decomposition work is in line with the application advocated.

5. Any pictorial representation or block diagram need to clearly figure out your whole work.

We have modified Fig. 1 in order to demonstrate the algorithmic architecture on the real-world data. We hope this improvement will more clearly show how the algorithms is used.

---

## Author Comment (AC2) · 27 Mar 2019

**Comments for REFEREE 2**

The paper entitled "Scalable Diagnostics and Data Compression for Global Atmospheric Chemistry using Ristretto Library" is well written. it is one of the needed research areas of current scenario. The authors are focusing to apply the dimensionality reduction for the probable compression of spatial features of the Global Atmospheric Chemistry data and for further analytics. The authors are requested to give the clarifications to the following queries for the possible acceptance of the paper.

1. The compression is achieved by means of dimensionality reduction. To achieve additional compression performance (Even the lossless entropy encoding) may be included in this proposed work to yield more compression ratio, this will even reduce the data size.

This type of compression is more of a straight-forward data-compression algorithm. The compression we are talking about here is in retaining a small number of modes which allow us to reconstruct the entire data matrix. This is an apples and oranges comparison. However, even when we compress to a small number of modes, these can be encoded then by computer science like data-compression algorithms. Thus this compression can work on top of our reduction.

However, to address this issue, we have taken the word compression out of the title, abstract and intro and referred to it in Section 5 in terms of how we consider compression. We hope this will remove confusion.

2. "Taking logarithm of the data" is mentioned in the paper, it would be better to give detail about this along with necessary references.

There are no references here, just the observation that if the data is not normalized in some way, then the only feature found is the giant spike in chemistry in China. This "normalization" procedure is common for PCA analysis where each row is normalized to be mean zero and unit variance so that different data types (some measured in large numbers and some in small numbers) can be equitably compared. This was the philosophy in taking the log of the data. By doing so, one can see the spatio-temporal feature beyond the large spike which would drown out all other signals. To address this, we propose adding the following additional sentences to 4.1

"Normalization of data is a common practice in data science. Indeed, the ubiquitous PCA analysis requires that each measurement type in the data have mean zero and unit variance. If this is not enforced, then those signals that are measured with large numbers will simply drown out the signals measured in small numbers. Thus the units
of the different measurements are neutralized by requiring a mean zero and unit variance. Similarly here, the large spike in the data is so large that the rest of the data is like noise comparatively. By normalizing with the logarithm, a more balanced global view of the chemistry dynamics can be extracted from the modal structures."

3. What is the reason for representing the Global Atmospheric Chemistry Data in 2D array in this proposed work? Why didn't considered as high dimensional data such as Tensor? Please justify the reasons.

Tensor decomposition are indeed possible with randomized methods. In fact, there are two recent papers (which we will cite in the revised version) that provide a nice architecture for this.

@article{erichson2017randomized, title={Randomized CP tensor decomposition}, author={Erichson, N Benjamin and Manohar, Krithika and Brunton, Steven L and Kutz, J Nathan}, journal={arXiv preprint arXiv:1703.09074}, year={2017} }

@article{battaglino2018practical, title={A practical randomized CP tensor decomposition}, author={Battaglino, Casey and Ballard, Grey and Kolda, Tamara G}, journal={SIAM Journal on Matrix Analysis and Applications}, volume={39}, number={2}, pages={876–901}, year={2018}, publisher={SIAM} }

To answer the question, these tensor decompositions could indeed by used. And we did try our randomized tensor decomposition on this data and found it did not provide much improvement and insight to the flattened data and standard SVD. Moreover, the standard randomized SVD method are simply faster and easier to use and have all the extra nuanced decompositions that can be easily added. Given there was no performance boost, we used the standard and robust data flattening method. Additionally, tensor decompositions, unlike SVD, are not unique and thus there is some variability in their ultimate output depending upon the specific optimization used.

In part, the lack of improvement is because the three directions (x, y and z) are all the

GMDD
same data measurements. For distinct data in each direction, i.e. pressure, temperature, moisture, for instance, the data flattening would be less helpful, and the tensor decomposition would be more appropriate.

At the end of the second paragraph of the conclusion, we propose adding the following:

"Randomized tensor decompositions~\cite{erichson2017randomized, battaglino2018practical} are also viable for producing scalable diagnostic features of the global chemistry data. However, for the specific data considered here, little or no improvement was achieved. However, in future work, we will consider such tensor decompositions across space, time and chemicals where the randomized tensor decomposition is ideally suited for extracting higher-dimensional features."

4. Mention, Among RSVD, NMF and SPCA, which is appropriate for your Global Atmospheric Chemistry Data analytics.

All these methods are appropriate for the data analytics on global chemistry data. We realize now that we should have been more specific about when and why one would use some of the various methods. We would propose adjusting he introductory part of Sec. 3 (just before 3.1) to the following: "The following subsections detail a probabilistic framework for matrix decompositions that includes a nonnegative matrix factorization as well as a sparsity-promoting technique. The mathematical architectures proposed provide scalable computational tools for the analysis of global chemistry dynamics. Moreover, by providing three different dimensionality architectures, a more nuanced objective analysis of the dominant spatio-temporal patterns that emerge in the global chemistry dynamics. The standard analysis would be a simple randomized SVD decomposition whereby the dominant correlated structures involves restricting the dominant spatio-temporal structures involves restricting the dominant spatio-temporal considerations. Specifically, the nonnegative matrix factorization restricts all chemicals to positive concentrations, a restriction which is physically motivated and especially important for diagnostics when

GMDD
physical interpretation is required. The randomized SVD will generally produce negative concentration of chemicals in individual modes, but the overall concentration is positive when the modes are summed together. Likewise, the sparse PCA analysis zeroes out very small concentrations so that the modes extracted highlight only nonzero contributions to the dynamics. This is an important modification of the randomized SVD since it generally produces all nonzero entries in the modal structures, regardless if it is physical. This is due to the least-square nature of the SVD algorithm. Again, a sparsification penalty produces modes where only the dominant correlations are nonzero. What one chooses to use may depend strongly on the application intended. Regardless, the suite of methods allows for a more nuanced view of the data."

5. You have specified ozone (O3) in your manuscript, is there any other key species are found in the atmosphere? If so, mention it in your manuscript.

Yes, there are several other chemicals that are critically important, and they have all been included in the supplemental material, including NO, NO2, OH, Isoprene, CO. To make this more clear, we would propose making a stronger statement at the beginning of Sec. 4. Specifically, we would change the beginning of Sec. 4 to the following:

"In this section we illustrate results from the decomposition of the GEOS-Chem model output using absolute concentration of ozone (O\$\_{3}\$) as an example. The supplementary material provides diagnostics for five additional chemicals known to dominate the global atmospheric chemistry dynamics. The additional five chemical species, including NO, NO\$\_2\$, OH, Isoprene (ISOP) and CO, are known to be equally important to ozone. For succinctness of the manuscript, we only present ozone here and the others in the supplement. Overall, there are close to two hundred chemicals that are interacting dynamically. Each chemical of interest can be diagnostic in a similar fashion to ozone in order to determine its dominant global variability. It remains an open research question how the interactions across the entire chemical space ultimately drive the observed variability. The scalable diagnostics advocated here provides a computational architecture allowing scientists to explore this further by providing global

GMDD
diagnostics for all chemicals in a computationally tractable manner."

6. It would be better to provide the Bit Rate of compressed data as well as original data in a tabular form.

To follow on the comments to point 1, we don't want to confuse "compression" from the algorithmic data compression point of view, with compression in storing a low-rank set of modes. We've highlighted here that only a small fraction of the modes need to computed in order to represent the entire large-scale data set, which has very little to do with bit rate compression. We have added the following to Section 5 to clarify this:

"Specifically, the compression advocated here is achieved by producing a low-rank representation for constructing the high-dimensional data, i.e. it should not be confused with standard data compression algorithms."